# Cytotoxic CX3CR1⁺ Vδ1 T cells clonally expand in an interplay of CMV, microbiota, and HIV-1 persistence in people on antiretroviral therapy

**Nived Collercandy**[1,2], **Camille Vellas**[1,3], **Manon Nayrac**[1¤], **Mary Requena**[1,3], **Thomas Richarme**[1], **Anne-Laure Iscache**[1], **Justine Latour**[3], **Karl Barange**[4], **Laurent Alric**[5,6], **Guillaume Martin-Blondel**[1,2,6], **Matteo Serino**[7], **Jacques Izopet**[1,3,6], **Pierre Delobel**[1,2,6]*

1 INSERM UMR 1291, CNRS UMR 5051, Université de Toulouse, Toulouse Institute for Infectious and Inflammatory Diseases, Toulouse, France, 2 CHU de Toulouse, Service des Maladies Infectieuses et Tropicales, Toulouse, France, 3 CHU de Toulouse, Laboratoire de Virologie, Toulouse, France, 4 CHU de Toulouse, Service d'Hépato-Gastro-Entérologie, Toulouse, France, 5 CHU de Toulouse, Service de Médecine Interne et Immunologie clinique, Toulouse, France, 6 Université de Toulouse, Toulouse, France, 7 Institut de Recherche en Santé Digestive, INSERM UMR 1220, INRAe, ENVT, Université de Toulouse, Toulouse, France

¤Current Address: Centre de Recherche du Centre Hospitalier Universitaire de Montréal, Université de Montréal, Canada

* delobel.p@chu-toulouse.fr

## Abstract

Vδ1 γδ T cells are key players in innate and adaptive immunity, particularly at mucosal interfaces such as the gut. An increase in circulating Vδ1 cells has long been observed in people with HIV-1, but remains poorly understood. We performed a comprehensive characterization of Vδ1 T cells in blood and duodenal intra-epithelial lymphocytes, obtained from endoscopic mucosal biopsies of 15 people with HIV-1 on antiretroviral therapy and 15 HIV-seronegative controls, in a substudy of the ANRS EP61 GALT study (NCT02906137). We deciphered the phenotype, functional profile, single-cell transcriptome and repertoire of Vδ1 cells and unraveled their relationships with the possible triggers involved, in particular CMV and microbiota. We also assessed whether Vδ1 T cells may play a role in controlling the HIV-1 reservoir. Vδ1 T cells were mainly terminally differentiated effectors that clonally expanded in the blood with some trafficking with the gut of people with HIV-1. Most expressed CX3CR1 and displayed a highly cytotoxic profile, but low cytokine production, supported by a transcriptomic shift towards enhanced effector lymphocytes. This expansion was associated with CMV status and markers of occult replication, but also with changes in the duodenal and blood-translocated microbiota. Cytotoxic, but not IFN-γ-producing, Vδ1 T cells were negatively associated with cell-associated HIV-1 RNA in both the blood and duodenal compartments. The increase in Vδ1 T cells observed in people with HIV-1 has multiple triggers, particularly CMV and microbiota, and may in turn contribute to the control of the HIV-1 reservoir.

**Data availability statement:** The sequence data supporting the findings of this study are available in the Sequence Read Archive (SRA) database under the identifiers PRJNA1293388 for γδ TCR repertoires, and PRJNA1284601 for blood and duodenum microbiota. The processed single-cell RNA sequencing data have been deposited in the Gene Expression Omnibus (GEO) database at the National Center for Biotechnology Information (NCBI) and are accessible through the GEO Series accession number GSE303439.

**Funding:** This work was supported by the French National Agency for Research on HIV/ AIDS and Emerging Infectious Diseases (ANRS-MIE https://anrs.fr/en/) (ECTZ204599 to PD; ECTZ205647 to NC) and Sidaction https:// www.sidaction.org (12937 to CV). ANRS MIE was the sponsor of the ANRS EP61 GALT study. ANRS-MIE and Sidaction provided grants to finance the project. The funders had no role in study design, data collection and analysis, decision to publish, or preparation of the manuscript.

**Competing interests:** The authors have declared that no competing interests exist.

## Author summary

Vδ1 T cells are key immune cells, particularly in the intestinal mucosa, which is a major site of HIV-1 persistence during antiretroviral therapy. Disruption of the gut microenvironment in people with HIV-1 is associated with chronic dysbiosis, increased bacterial translocation, and subsequent immune activation and systemic inflammation. An increase in circulating Vδ1 T cells has long been observed in this setting, regardless of the stage of infection, and persists even with effective antiretroviral therapy, but the reasons for this remain poorly understood.

Our findings revealed the interplay between CMV, microbiota, and HIV-1 and Vδ1 T cell immune responses, particularly in the gut, to explain the expansion of Vδ1 T cells observed in people with HIV-1. They also highlight the role of cytotoxic Vδ1 T cells in targeting residual HIV-1-infected cells. This may lead to considering Vδ1 T cell-based immunotherapy strategies to target the HIV-1 reservoir.

## Introduction

The intestinal mucosa appears to be a key player in the pathophysiology of HIV-1, both as a direct target for HIV-1 replication and as a source of chronic antigenic stimulation from multiple sources in people with HIV-1 (PLWH). Indeed, the intestinal mucosa is highly enriched in CCR5⁺CD4⁺ T cells, which are specifically targeted and depleted during the early stages of HIV-1 infection. This leads to disruption of the intestinal immune barrier [1,2]. Even on antiretroviral therapy (ART), immune reconstitution of the intestinal mucosa remains incomplete [3,4]. Disruption of the gut microenvironment is associated with chronic dysbiosis, increased bacterial translocation, and subsequent immune activation and systemic inflammation in PLWH [5,6]. The gut is also a major site of HIV-1 persistence during ART, with enrichment of intact proviral DNA, which is the form responsible for viral rebound upon ART interruption. Persistent transcription from HIV-1 proviruses, both intact and defective, also appears to be a source of residual viral antigen production [7–10]. In addition, human cytomegalovirus (CMV) can also reactivate in the intestinal mucosa of PLWH and may be a major source of T-cell activation [11]. These multiple and persistent antigenic stimuli may contribute to chronic immune activation and exhaustion in PLWH [12].

Each intestinal segment has specific characteristics in antigen uptake and immune cell composition [13,14]. Intra-epithelial lymphocytes (IEL) are more abundant in the proximal intestine, particularly in the duodenum. Due to their anatomical location, duodenal IEL are at the forefront of the intestinal barrier defense against a wide range of digestive pathogens and dietary antigens, including bacteria, viruses, and fungi, which in turn expand the IEL [15]. IEL are divided into conventional αβTCR T cells, mostly CD8⁺, and unconventional T cells, of which γδ T cells are an important subset.

Vδ$_1$ T cells are the more predominant subsets of γδ T cells in the intestinal epithelium [16]. They are involved in both innate and adaptive immune responses

[17]. While they can be triggered in a TCR-independent manner by natural killer receptor-mediated recognition of stress ligands, particularly in epithelia, they can also recognize conserved patterns in a major histocompatibility complex (MHC)-unrestricted manner through their TCR [16,18]. Remarkably, their repertoire can be increasingly shaped with age by exposure to various pathogens, most notably CMV, but also malaria or EBV, with pathogen-driven clonal expansion [19–23]. Expanded $V\delta_1$ T cells exhibit cytotoxic effector properties and express the chemokine receptor CX3CR1 [19]. Expanded clonotypes can be found in peripheral tissues as well as in the circulation, although $V\delta_1$ T cells are rare in the blood, where the most abundant subset of γδ T cells is classically the $V\delta_2$ subset [24]. In tissues, in addition to their role against pathogens, $V\delta_1$ IEL may also have anti-tumor properties [25].

Surprisingly, in the setting of HIV-1 infection, CD27⁻CD45RA⁺ $V\delta_1$ T cells expand in the circulation, and outnumber the $V\delta_2$ subset, both in primary and chronic HIV-1 infection, and even under ART [26–29]. The $V\delta_1/V\delta_2$ ratio inversion seems to occur only at the chronic stage. This has been observed during both HIV-1 and SIV infection [30,31]. The evolution of γδ T cell numbers among IEL is less clear and may depend on the stage of HIV-1 infection [27,32]. The expansion of $V\delta_1$ cells in the blood of PLWH, a subset that is more abundant in the mucosa, raises the question of their intestinal origin. However, the trigger for the expansion of $V\delta_1$ T cells and the inversion of the $V\delta_1/V\delta_2$ ratio in PLWH is not clearly understood. In SIVsm-infected rhesus macaques, the increase in $V\delta_1$ T cells has been correlated with microbial translocation [33]. HIV-1 may also be an inducer and potentially a target of $V\delta_1$ T cells, which may have antiviral activity against HIV-1, particularly those bearing the activator receptor NKG2C, and thus may contribute to the control of HIV-1 replication [28,34,35].

Therefore, we performed a comprehensive characterization of $V\delta_1$ T cells in blood and duodenal IEL obtained from endoscopic mucosal biopsies in PLWH on ART and uninfected controls. We deciphered their phenotype, functions, transcriptome and repertoire and unraveled their relationships with CMV, microbiota and the HIV-1 reservoir.

Here, we found that $V\delta_1$ T cell expansion in PLWH is due to clonal expansion of terminally differentiated effector memory cells that express the chemokine receptor CX3CR1 and exhibit highly cytotoxic functions supported by expression of effector pathways. Some expanded $V\delta_1$ T cells show features of intestinal recirculation. Importantly, this expansion of CX3CR1⁺ $V\delta_1$ T cells was found specifically in PLWH with positive CMV serologic status. We observed associations between markers of occult CMV replication, changes in the duodenal and blood microbiota, and the phenotype of $V\delta_1$ T cells. Finally, cytotoxic $V\delta_1$ T cells negatively correlated with residual HIV-1 RNA in both blood and duodenum, suggesting that these cells may contribute to HIV-1 control.

## Results

### Study population

This study was nested within the ANRS EP61 GALT study, which enrolled 42 PLWH on ART and HIV seronegative controls. Their full clinical and virological details have been previously published [7]. Here, we performed a sub-study in which the main population consisted of 15 PLWH on ART and 15 HIV seronegative controls, all seropositive for CMV IgG antibodies, with available duodenal biopsies and PBMC (Fig 1A; S1 Table). The median age was 54 years for PLWH (80% men) and 58 years for controls (73% men). Included subjects were PLWH who initiated ART in the chronic stage of infection and had sustained virologic suppression (plasma HIV-1 RNA < 30 copies/mL) for a median of 9 years [IQR, 5–14], and who had a median CD4 T-cell count of 812 cells/µL [IQR, 590–956] at the time of sampling. In addition, we studied CMV-seronegative individuals with available PBMC, 12 of whom were PLWH on ART with clinical characteristics close to those of CMV-positive PLWH, and 12 of whom were HIV-seronegative controls matched for age and sex (S1 Table).

### The increased proportion of $V\delta_1$ T cells in the blood of PLWH is mainly due to the expansion of CX3CR1⁺ TEMRA cells

The phenotype of the $V\delta_1$ and $V\delta_2$ subsets of γδ T cells was assessed by flow cytometry in both PBMC and duodenal IEL from each individual in the main CMV-seropositive study population (Figs 1B and S1).

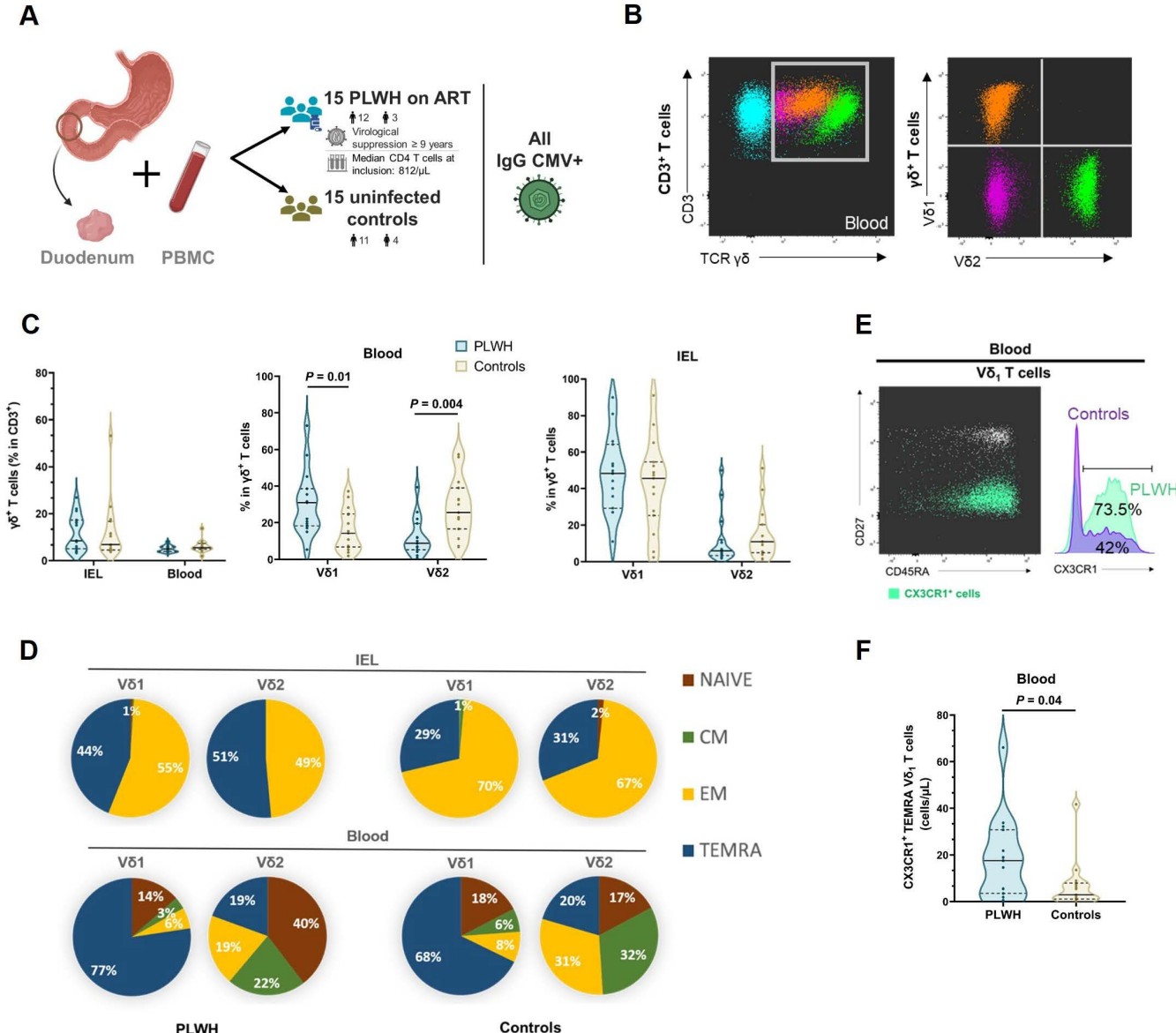

**Fig 1. Increased frequency of Vδ₁ T cells in the blood of PLWH is associated with the expansion of cells with a CX3CR1⁺ TEMRA phenotype in the blood.** (A) Schematic representation of sample origin (created with Biorender). Duodenal endoscopic biopsies (n = 8) were either stored in RNA-later for subsequent sequential lysis and viral and bacterial RNA and DNA extraction, or processed and cryopreserved for subsequent analysis of IEL. (B) Flow cytometric analysis of γδ⁺ T cells among CD3⁺ T cells and Vδ₁ and Vδ₂ subsets in a PLWH blood sample. (C) Violin plots of the frequencies of γδ⁺ T cells among CD3⁺ T cells, and their Vδ₁ and Vδ₂ subsets in IEL and blood from PLWH and HIV seronegative controls (n = 15 PLWH and 15 controls, all CMV seropositive). Comparisons were made using Welch's t-test. The graphs show the median (solid bar) and the first and third quartiles (dashed bars). (D) Proportions of naive (CD27⁺CD45RA⁺), central memory (CM; CD27⁺CD45RA⁻), effector memory (EM; CD27⁻CD45RA⁻), and terminally differentiated effector memory (TEMRA; CD27⁻CD45RA⁺) of the Vδ₁ and Vδ₂ subsets in IEL and blood of PLWH and HIV seronegative controls. (E) Representative dot plot of CX3CR1 expression among Vδ₁ T cells in PLWH blood according to their CD27 and CD45RA expression, and histogram of the median percentage of CX3CR1⁺ among circulating Vδ₁ T cells. (F) Violin plots of TEMRA CX3CR1⁺ Vδ₁ T cell counts in blood (n = 15 PLWH and 15 HIV seronegative controls, all CMV seropositive). Comparison was made using Welch's t-test.

As expected, γδ T cells were enriched in the duodenal mucosa and consisted mainly of $V\delta_1$ cells. We observed no difference between PLWH and controls in the frequency of total γδ T cells among $CD3^+$ T cells in both PBMC and IEL, nor in the frequencies of $V\delta_1$ and $V\delta_2$ cells among γδ T IEL (Fig 1C), nor in the absolute numbers of $V\delta_1$ and $V\delta_2$ (S2 Fig) on microscopic analysis of duodenal biopsies. We confirmed the previously reported inversion of the $V\delta_1/V\delta_2$ ratio in PLWH blood with an increased frequency of the $V\delta_1$ subset (median, 31% [IQR, 18.9-37.1]) in PLWH compared to controls (median, 14.3% [IQR, 7-24.6]; $P=0.01$, Welch's t-test), accompanied by a decrease in the $V\delta_2$ subset (median 8.8% [IQR, 5.9-15.8] for PLWH, and 25.6% [IQR, 16.7-35.9] for controls; $P=0.004$, Welch's t-test) (Fig 1C). The differentiation phenotype of the $V\delta_1$ and $V\delta_2$ subsets showed distinct populations in the duodenal epithelium compared to the circulation (Fig 1D). IEL were almost exclusively either $CD27^-CD45RA^-$ effector memory cells or $CD27^-CD45RA^+$ terminally differentiated effector memory (TEMRA) cells. $V\delta_1$ T cells from PLWH showed an increased frequency of TEMRA cells in both compartments, particularly in the blood where they represented the majority of cells regardless of HIV status. Circulating effector $V\delta_1$ T cells were characterized by the expression of CX3CR1, which is predominantly expressed on the TEMRA subset. $CX3CR1^+$ cells were highly enriched in circulating $V\delta_1$ T cells of PLWH (median 73.5% [IQR, 44-79.5] vs. 42% in controls [IQR, 26.4-67.1], $P=0.11$, Welch's test) (Fig 1E). By contrast, CXCR3 tended to be less expressed in circulating $V\delta_1$ T cells of PLWH than in controls, while CCR6 and CCR9 were expressed at lower levels, with no difference in frequency between the two groups (S3 Fig). The number of circulating $CX3CR1^+$ $V\delta_1$ TEMRA cells was increased in PLWH compared to controls (median 17.6 cells/μL [IQR, 4.5-26.3] vs. 2.9 cells/μL [IQR, 1.2-7.5], respectively, $P=0.04$, Welch's test; Fig 1F). Clustering of circulating $V\delta_1$ T cells using the unsupervised opt-SNE algorithm delineated distinct effector cell phenotypes (Fig 2A). Cluster 5, which included cells with a TEMRA phenotype expressing CX3CR1, the activator receptor NKG2C, and the activation and exhaustion marker TIM-3, was significantly more frequent in PLWH (median 21% [IQR, 3.7-40.6]) compared to controls (median 7.6% [IQR, 3-14.5]; $P=0.04$, Welch's t-test). Expression of TIM-3 and NKG2C was increased on blood $CX3CR1^+$ $V\delta_1$ T cells from PLWH compared to controls (Fig 2B). Expression of exhaustion markers and especially their co-expression was increased on TEMRA $V\delta_1$ T cells from PLWH compared to controls (S4 Fig).

Compared to circulating $V\delta_1$ T cells, duodenal $V\delta_1$ IEL were globally characterized by the almost ubiquitous expression of the β7 integrin, CD103 (αE) and CD69 and higher expression levels of NKG2C and TIM-3 but lower expression of CX3CR1 (Fig 2C). Compared to controls, $V\delta_1$ IEL from PLWH showed higher expression of NKG2C (Fig 2D). The level of NKG2C expression in duodenal $V\delta_1$ IEL was positively correlated with the inversion of the $V\delta_1/V\delta_2$ ratio observed in the blood (n=30, Spearman's r=0.39, $P=0.03$; Fig 2E). Circulating $CX3CR1^+$ $V\delta_1$ T cells expressing the gut-homing markers β7 and CD103 were increased in PLWH compared to controls (median 1.1/μL [IQR, 0.6-2.6] vs. 0.2/μL [IQR, 0-0.7] in PLWH and controls, respectively, $P=0.01$, Welch's t-test; Fig 2F), suggesting recirculation of effector $V\delta_1$ T cells from the gut.

## Circulating $CX3CR1^+$ TEMRA $V\delta_1$ T cells have a highly cytotoxic profile but produce fewer pro-inflammatory cytokines

We further investigated the functions of the Vδ1 T cells by intracellular flow cytometry staining. We evaluated cytotoxic effector functions using unstimulated Granzyme B (GzmB) and Perforin staining, and pro-inflammatory cytokine production using PMA/ionomycin-stimulated Interferon-γ (IFN-γ) and Tumor Necrosis Factor α (TNF-α) staining in both circulating and IEL $V\delta_1$ T cells (Fig 3A, B). Circulating TEMRA $V\delta_1$ T cells exhibit a cytotoxic phenotype with high production of GzmB and Perforin, particularly for $CX3CR1^+$ TEMRA $V\delta_1$ (median of double positive: 85% [IQR: 71–94]) in both PLWH and controls. In contrast, circulating $CX3CR1^+$ $V\delta_1$ T cells had the lowest production of IFN-γ and TNF-α compared to $CX3CR1^-$ cells. Duodenal $CX3CR1^+$ $V\delta_1$ IEL showed much lower GzmB, Perforin, IFN-γ and TNF-α production compared to circulating cells. However, although lower than in their circulating counterparts, Perforin expression was higher in $V\delta_1$ IEL from PLWH compared to controls (median of 26% [IQR, 20–35] vs. 10% [IQR, 9–16] $GzmB^-/Perforin^+$ cells, in PLWH vs. controls, respectively; $P<0.01$, Welch's t-test; S5 Fig). To further explore the functional profile of $V\delta_1$ T cells, we

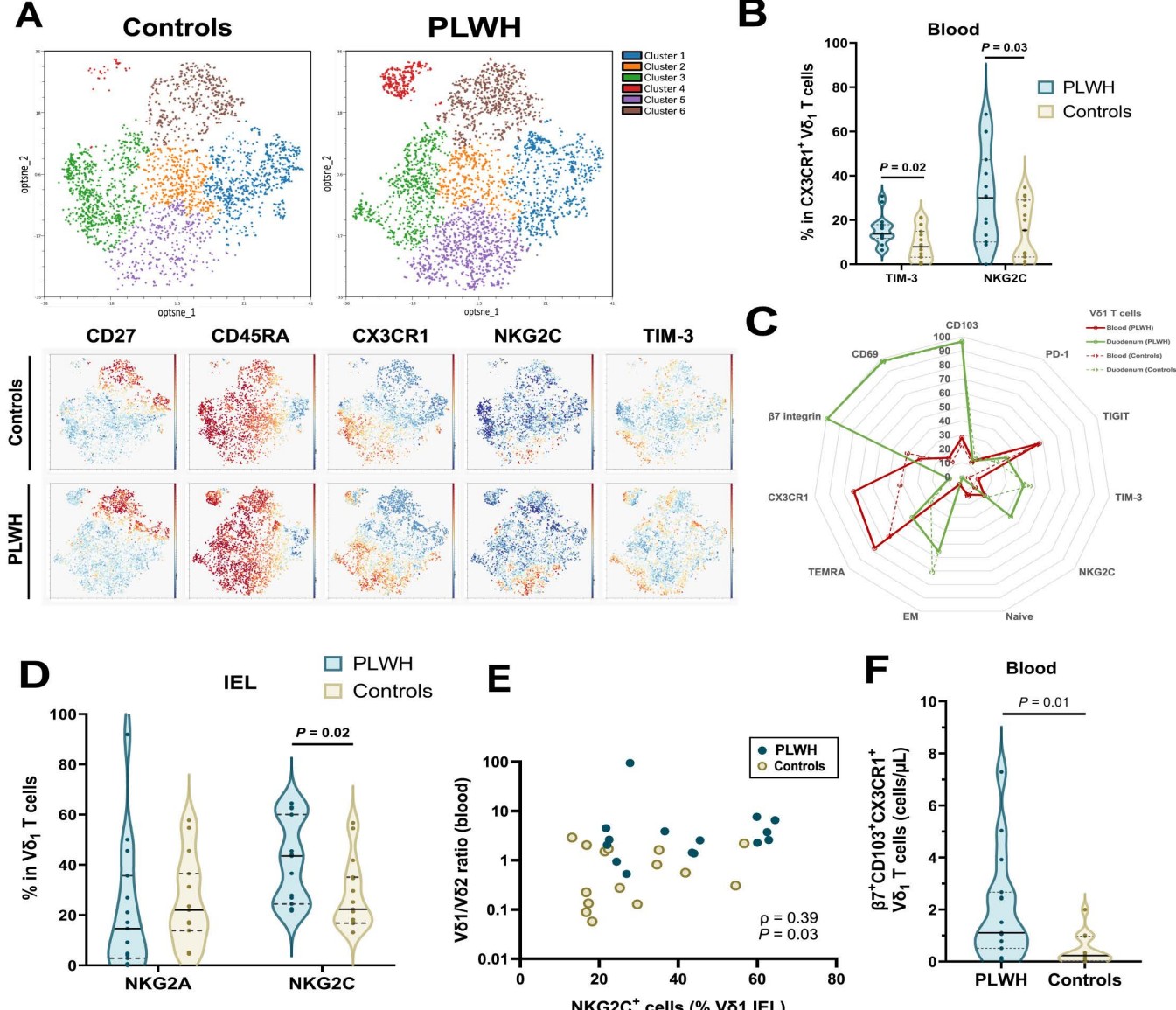

**Fig 2. Vδ₁ T cells of PLWH present higher expression of NKG2C in both the blood and the duodenum.** (A) Opt-SNE representation of Vδ₁ T cell clustering obtained from flow cytometric data. Clustering was performed on a total of n = 15 PLWH and n = 15 HIV seronegative controls combined, all CMV seropositive. Vδ₁ T cell counts were equalized between individuals after a subsampling step. Expression of CD27, CD45, CX3CR1, NKG2C and TIM-3 in the clusters is shown with a color gradient (each marker has a different color scale to optimize the gradient visualization). (B) Violin plots of the frequencies of TIM-3⁺ and NKG2C⁺ cells among CX3CR1⁺ Vδ₁⁺ T cells in blood (n = 15 PLWH and 15 HIV seronegative controls, all CMV seropositive). Comparisons were made using the Welch's t-test. (C) Radar plot of phenotypic markers distinguishing blood and duodenal epithelial Vδ₁ T cells. Values are median percentages of expression of each marker among Vδ₁ T cells. (D) Violin plots of the frequencies of NKG2A and NKG2C among Vδ₁⁺ T cells in the IEL (n = 15 PLWH and 15 HIV seronegative controls, all CMV seropositive). Comparisons were made using Welch's t-test. (E) Positive correlation between blood Vδ₁/Vδ₂ ratio and NKG2C expression in IEL Vδ₁⁺ T cells (n = 15 PLWH and 15 HIV seronegative controls, all CMV seropositive). Spearman's correlation coefficient ($r_s$) and P-value are shown. (F) Violin plots of blood counts of β7⁺ CD103⁺ CX3CR1⁺ Vδ₁ T cells (n = 15 PLWH and 15 HIV seronegative controls, all CMV seropositive). Comparison was made using the Welch's t-test. See also S4 Fig for the median co-expression of activation/exhaustion markers PD-1, TIM-3 and TIGIT.

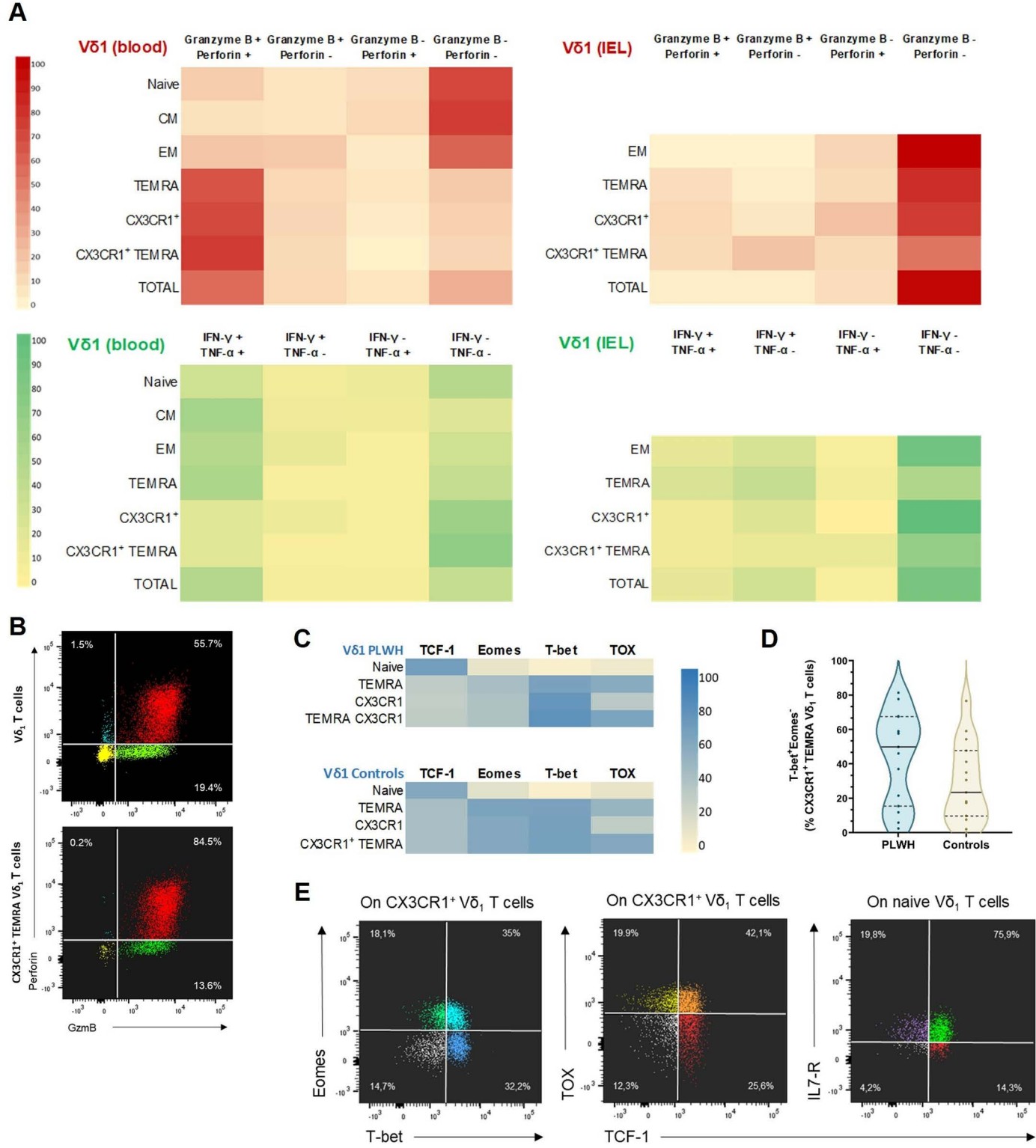

**Fig 3. Circulating CX3CR1+ TEMRA Vδ₁ T cells have a highly cytotoxic profile but produce few pro-inflammatory cytokines.** (A) Heatmaps of the mean percentage expression of Granzyme B, Perforin, IFN-γ and TNF-α among Vδ1 T cells. The red and green heatmaps are the results of 2 different flow cytometry panels. The same panels were used for both PBMC and IEL staining. Blood staining was performed on n = 15 PLWH and 15

HIV seronegative controls, all CMV seropositive. IEL staining was performed on n = 5 PLWH and n = 5 HIV seronegative controls, all CMV seropositive. Results shown are the mean of PLWH and HIV seronegative controls combined. (B) Representative dot plots of Granzyme B (GzmB) and Perforin expression in V$\delta_1$ T cells and CX3CR1$^+$ TEMRA V$\delta_1$ T cells. (C) Heatmaps of the median percentage expression of the transcription factors TCF-1, Eomes, T-bet and TOX among blood V$\delta$1 T cells. Results are from a single flow cytometry panel. Blood staining was performed on n = 15 PLWH and n = 15 HIV seronegative controls, all CMV seropositive. (D) Violin plots of CX3CR1$^+$ TEMRA T-bet$^+$ Eomes$^-$ V$\delta_1$ T cell percentages in blood (n = 15 PLWH and 15 HIV seronegative controls, all CMV seropositive). (E) Representative dot plots of Eomes, T-bet, TOX and TCF-1 expression in blood CX3CR1$^+$ V$\delta_1$ T cells and naive (CD27$^+$CD45RA$^+$) V$\delta_1$ T cells. See also S5 Fig for duodenal cytotoxic function.

examined the expression of key transcription factors TCF-1, TOX, T-bet, and Eomes in the blood using intra-nuclear flow cytometry staining (Fig 3C–E). Effector CX3CR1$^+$ V$\delta_1$ T cells showed higher expression of T-bet and lower expression of Eomes than other TEMRA V$\delta_1$ T cells, particularly in PLWH.

## Circulating V$\delta_1$ T cell transcriptome shifts toward an enhanced cytotoxic effector profile

Single-cell RNA sequencing (scRNAseq) was performed on sorted circulating V$\delta_1$ T cells from 6 PLWH and 6 controls to characterize the transcriptional reprogramming that supports the enhanced effector phenotype of V$\delta_1$ T cells observed in PLWH. An average of 655 single V$\delta_1$ T cells per donor were sequenced. Unsupervised analysis revealed clustering of 6 profiles that differed between PLWH and controls (Fig 4A–C). Clusters 0 and 1 were more abundant in PLWH, whereas clusters 2–5 were more abundant in controls. Clusters 3–5 showed an early differentiation profile, with clusters 3 and 5 corresponding to naive T cells expressing high levels of *CD27*, higher levels of *CD28* and *CCR7*, and an important self-renewal potential with the expression of *IL7R, LTB, TCF7, LEF1* (Fig 4D, E). Cluster 4 had preserved but decreased *CD27* expression, and also showed features of effector cells with the expression of cytotoxicity-associated genes (especially *GZMK*, but also *GZMA, GZMH, PRF1*). In contrast, clusters 0–2 presented the most terminally differentiated profile with loss of *CD27, CD28, CCR7* expression and the highest level of *CX3CR1* expression. They expressed high levels of genes associated with cytotoxicity such as *GZMA, GZMB, GZMH, PRF1, GNLY*, the transcription factor *TBX21* and the NK family activator receptor NKG2C gene *NKG7*. Clusters 2 and 4 had higher expression of the pro-inflammatory cytokine gene *IFNG*, which was very low in the other effector-like clusters. Cluster 1 differed from clusters 0 and 2 mainly by a lower expression of the transcription factors *TOX*, associated with immune exhaustion, and of *STAT4* expression. Comparisons of gene expression within each cluster between PLWH and controls revealed a global increase in *FGL2* expression in PLWH, in addition to increased expression of *FNDC3B, DOK6, FRY,* which are involved in cell proliferation signaling in clusters 0 and 1, of *CD226* (DNAM-1 gene, associated with cell killing pathway) expression in cluster 1, and of *CX3CR1* and *S1PR5* (downregulated gene in resident memory T cells) expression in cluster 3 (S6 Fig). Terminally differentiated clusters co-expressed *ITGAE* and *ITGB7*, encoding CD103 and β7 integrins (Fig 4E), suggesting recirculation of effector V$\delta_1$ T cells from the gut.

## Circulating V$\delta_1$ T cells are clonally expanded in PLWH

To assess the clonality and diversity of the expanded V$\delta_1$ TCR repertoire, we performed dimer-avoided multiplex (dam)-PCR and NGS to analyze the CDR3 sequences of the hTRDV chain on sorted blood V$\delta_1$ T cells from 5 PLWH and 5 controls, and unsorted duodenal IEL (Fig 5A). Sequenced cells from the blood almost exclusively expressed the TRDV1 gene in accordance with the realized sorting. TRDV1 expressing cells accounted for an average of 54% of the duodenal epithelial γδ T cells, and TRDV1 clones also accounted for 54% of the sequenced clonotypes. The number of sequenced V$\delta_1$ T cells was higher in the PLWH blood samples (mean 5,253 [range 1,461–12,210] for PLWH, and 939 [204-2,001] for controls in blood; 565 [262–824] in PLWH duodenum). CDR3 analysis revealed a clonally focused repertoire in all subjects, although the clonal expansion appeared to be more pronounced in the blood of PLWH. Notably, for one of the 5 PLWH (#8547), the predominant circulating clone (45% of the sequenced V$\delta_1$ cells) was also found in the duodenal IEL

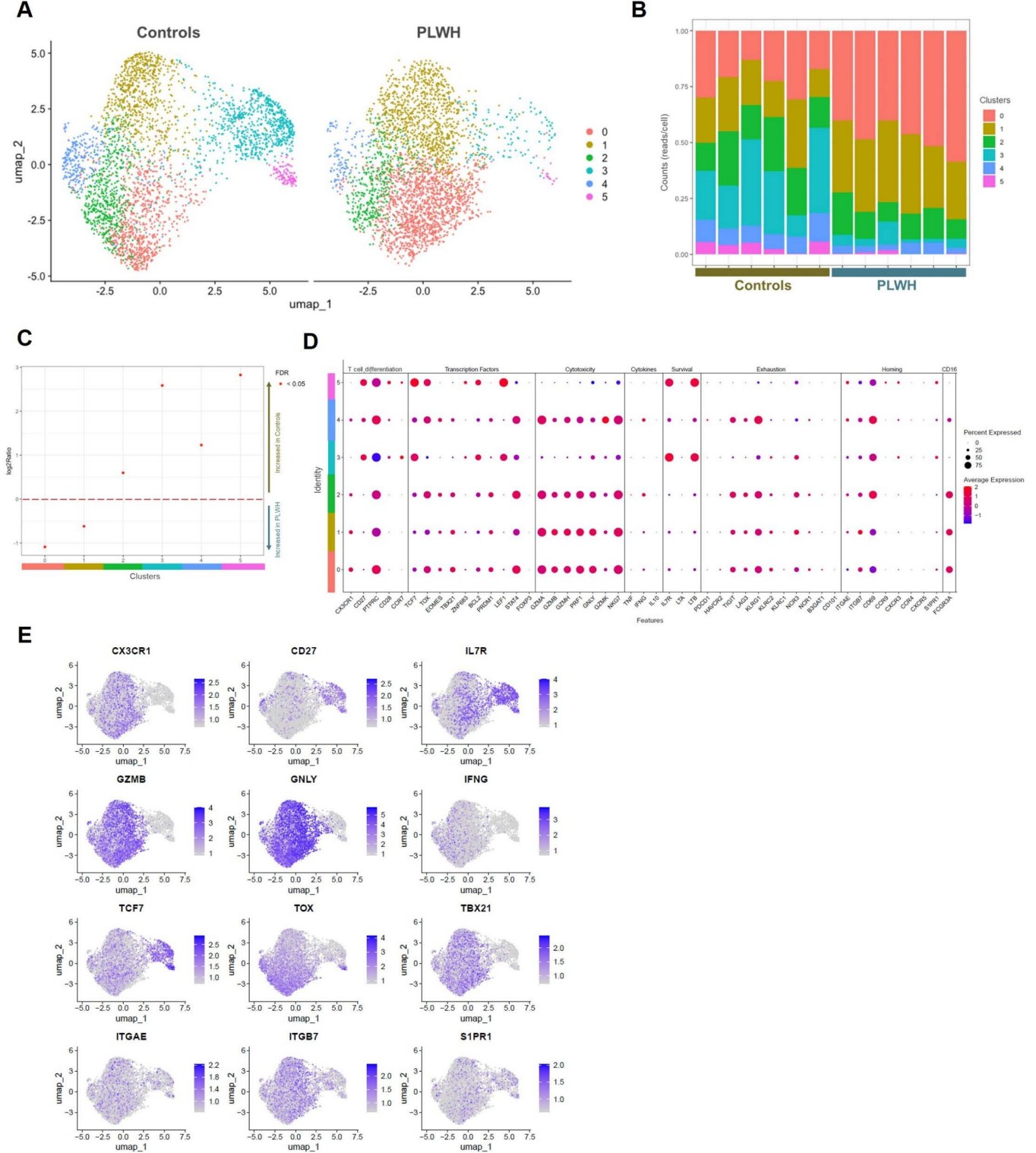

**Fig 4. Circulating Vδ₁ T cell transcriptome shifts toward an enhanced cytotoxic effector profile.** Data shown are the result of a single-cell RNA sequencing experiment performed on n = 7,865 sorted Vδ₁ T cells from the blood of 6 PLWH and 6 HIV seronegative controls, all CMV seropositive. (A) UMAP representation of single Vδ₁ T cells according to their transcriptomic profile, which allowed to distinguish 6 clusters of interest. (B) Proportion of

each cluster for each individual donor. (C) Log 2 ratio of each cluster according to its status (PLWH versus HIV seronegative control). A positive value was associated with an increased frequency in HIV seronegative controls compared to PLWH, and a negative value was associated with an increased frequency in PLWH compared to HIV seronegative controls. All log 2 ratios were significantly associated with an increased frequency of the corresponding cluster in either PLWH or HIV seronegative controls (false discovery rate (FDR) < 0.05). (D) Dot plot of the expression of genes of interest according to each cluster. Each dot size represents the percentage of cells expressing the corresponding gene within the cluster, and the color range indicates the average scaled expression level of the gene. (E) UMAP representation of the given gene expression level. The intensity of the purple color represents the level of RNA expression (quantified by the number of reads per cell for each gene) for the chemokine receptor CX3CR1, differentiation marker CD27, IL-7 receptor (CD127), cytotoxic effectors GzmB, granulysin, cytokine IFN-γ, transcription factors TCF7 (TCF-1), TOX, TBX21 (T-bet), homing integrins αE (CD103) and β7, and sphingosine-1-phosphate receptor (S1PR1). See also S6 Fig for volcano plots of the differentially expressed genes between PLWH and HIV seronegative controls.

of the same subject, suggesting recirculation between these 2 compartments. We used a diversity index (defined as 100 minus the area under the curve between the percentage of total reads and the percentage of unique CDR3s, when unique CDR3s are sorted by frequency from largest to smallest) and Hill's numbers (for which increasing values of q make the diversity measure progressively less sensitive to rare haplotypes) to assess the diversity of the TRDV1 clonotypes (Fig 5B, C). Rényi's plots of Hill's numbers were used to rank samples according to their degree of genetic complexity, with the highest curves corresponding to the most complex samples. TRDV1 diversity was lower in PLWH blood, compared to both control blood and matched duodenal IEL, and remained lower for all values of q (representing the order of the diversity). One control (#3124) presented a very low count of V$\delta_1$ T cells in blood (so only n = 204 V$\delta_1$ T cells were sequenced), leading to the highest diversity index (42.5), the main clone accounting for only 18% of the sequenced repertoire.

## CMV is associated with V$\delta_1$ T cells expansion in PLWH on ART

Since CMV has been described as a trigger of V$\delta_1$ T cell clonal expansion, we investigated the potential role of CMV co-infection in PLWH in the decreased diversity of the V$\delta_1$ repertoire. The flow cytometric phenotyping results of circulating V$\delta_1$ T cells obtained from the 15 CMV$^+$ PLWH and 15 CMV$^+$ controls were compared with 12 CMV$^-$ PLWH and 12 CMV$^-$ controls. There was an increase in the number of circulating γδ T cells in CMV-seropositive individuals, especially in PLWH (median of 85.2/μL in HIV$^+$CMV$^+$ vs. 39.4/μL in HIV$^+$CMV$^-$; $P = 0.01$, Welch's t-test) (Fig 6A). This was mainly related to the increase in the number and frequency of V$\delta_1$ T cells in the same groups (median of 24.7/μL in HIV$^+$CMV$^+$ vs. 4.5/μL in HIV$^+$CMV$^-$; $P = 0.001$, Welch's t-test), while the number of V$\delta_2$ T cells remained unchanged and their frequency among total γδ T cells decreased in HIV$^+$CMV$^+$ individuals. An inversion of the V$\delta_1$/V$\delta_2$ ratio was observed only in the HIV$^+$CMV$^+$ group with a median of 2.6 (IQR, 1.7-4.2), whereas the median ratio was < 1 in all other 3 groups (0.6 for HIV$^-$CMV$^+$ and HIV$^+$CMV$^-$, 0.5 for HIV$^-$CMV$^-$) (Fig 6B). Notably, the frequency of CX3CR1$^+$ cells among V$\delta_1$ T cells was highly dependent on CMV status (median of 73.5% in HIV$^+$CMV$^+$ vs. 27.6% in HIV$^+$CMV$^-$; $P = 0.001$, Welch's t-test) (Fig 6C). CMV viremia assessed by qPCR was negative in all CMV-seropositive subjects. The CMV IgG index was used as a surrogate for occult CMV replication and was significantly higher in PLWH compared to controls (median 407.5 vs. 156.3 AU/mL; $P = 0.025$, Welch's t-test) (Fig 6D).

## PLWH show changes in the duodenal mucosal and blood-translocated microbiota

Because of the anatomical interface between the IEL and the gut microbiota and the key role of the IEL in host gut barrier defense, we investigated the potential shift in duodenal mucosal microbiota and blood-translocated microbiota populations in PLWH in search of a trigger for mucosal stimulation and potential recirculation of V$\delta_1$ T cells. Duodenal biopsies from 24 PLWH and 25 controls, and blood samples from 33 PLWH and 39 controls, from the EP61 GALT study were used to perform pan-bacterial 16S ribosomal RNA PCR and NGS to identify and classify bacterial phyla, classes, orders, families, genera, and species. Overall microbiota diversity, as assessed by multiple indices, showed a higher diversity in the blood and duodenal mucosa of PLWH compared to controls (S7 Fig). The duodenal microbiota showed a higher abundance of

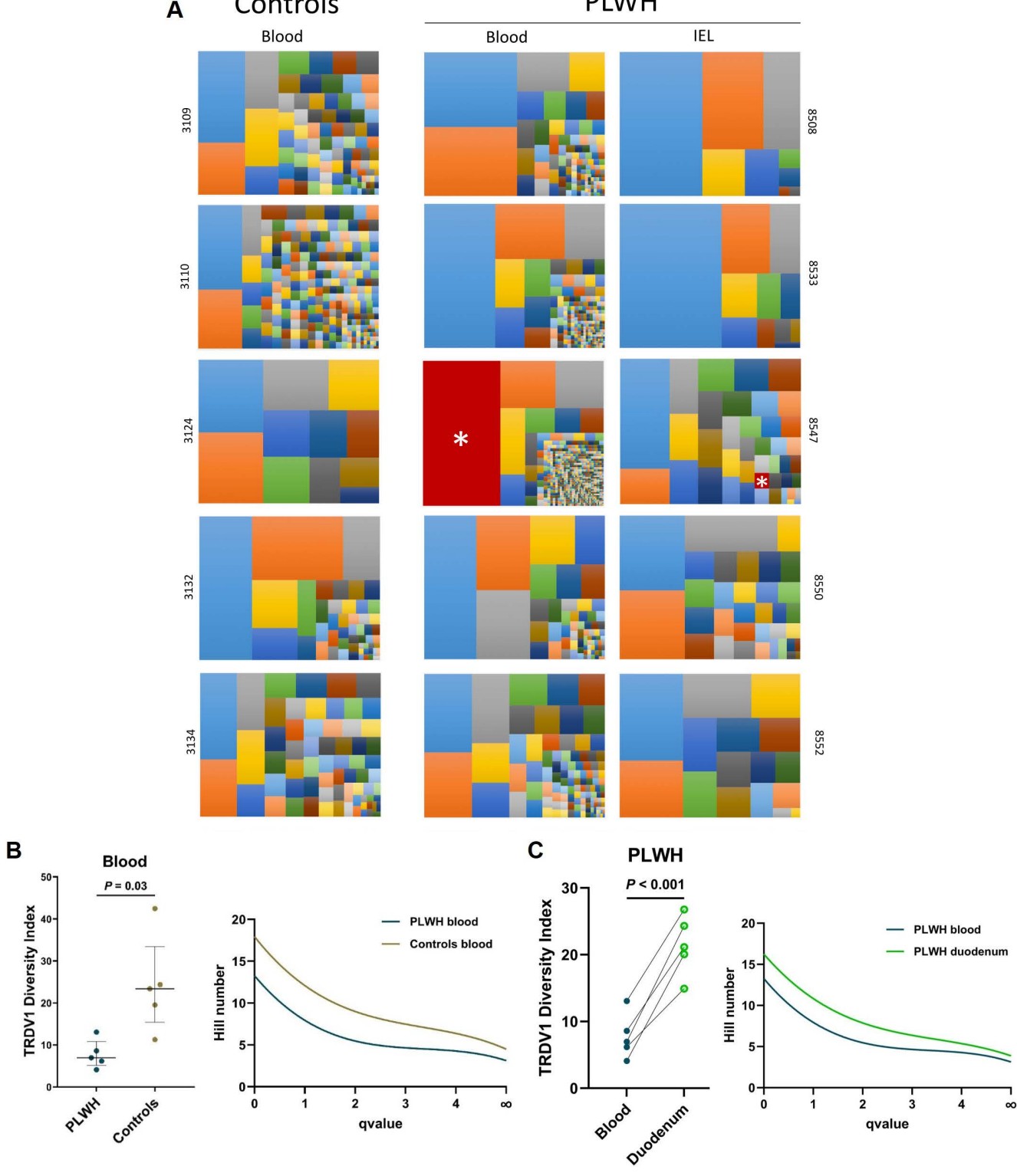

**Fig 5. Circulating Vδ₁ T cells are clonally expanded in PLWH.** Unsorted duodenal IEL and sorted Vδ₁ T cells from PBMC of the same 5 PLWH and sorted Vδ₁ T cells from PBMC of 5 HIV seronegative controls, all CMV seropositive, were used to sequence the TRDV1 chain repertoire. (A) Tree maps of unique CDR3 clonotypes. Each CDR3 color is randomly chosen and does not match between plots, except for the clonotype marked with a white

asterisk and colored in red, which represents an identical clonotype. Each colored square size represents the proportion of the clonotype within the total TRDV1 chain CDR3 repertoire. (B) Diversity index (function of the frequency of each CDR3 and the total number of unique CDR3s) for the TRDV1 chain in blood and Rényi's plot of mean Hill's numbers of order q = 0 to q = ∞, showing lower diversity in blood from PLWH (n = 5) than in HIV seronegative controls (n = 5). *P*-values are the result of Welch's t-test. (C) Diversity index for TRDV1 chain in the duodenal IEL and Rényi's plot of mean Hill's numbers of order q = 0 to q = ∞, showing higher diversity in duodenum (n = 5) than in blood of PLWH (n = 5). *P*-values are the result of paired t-test.

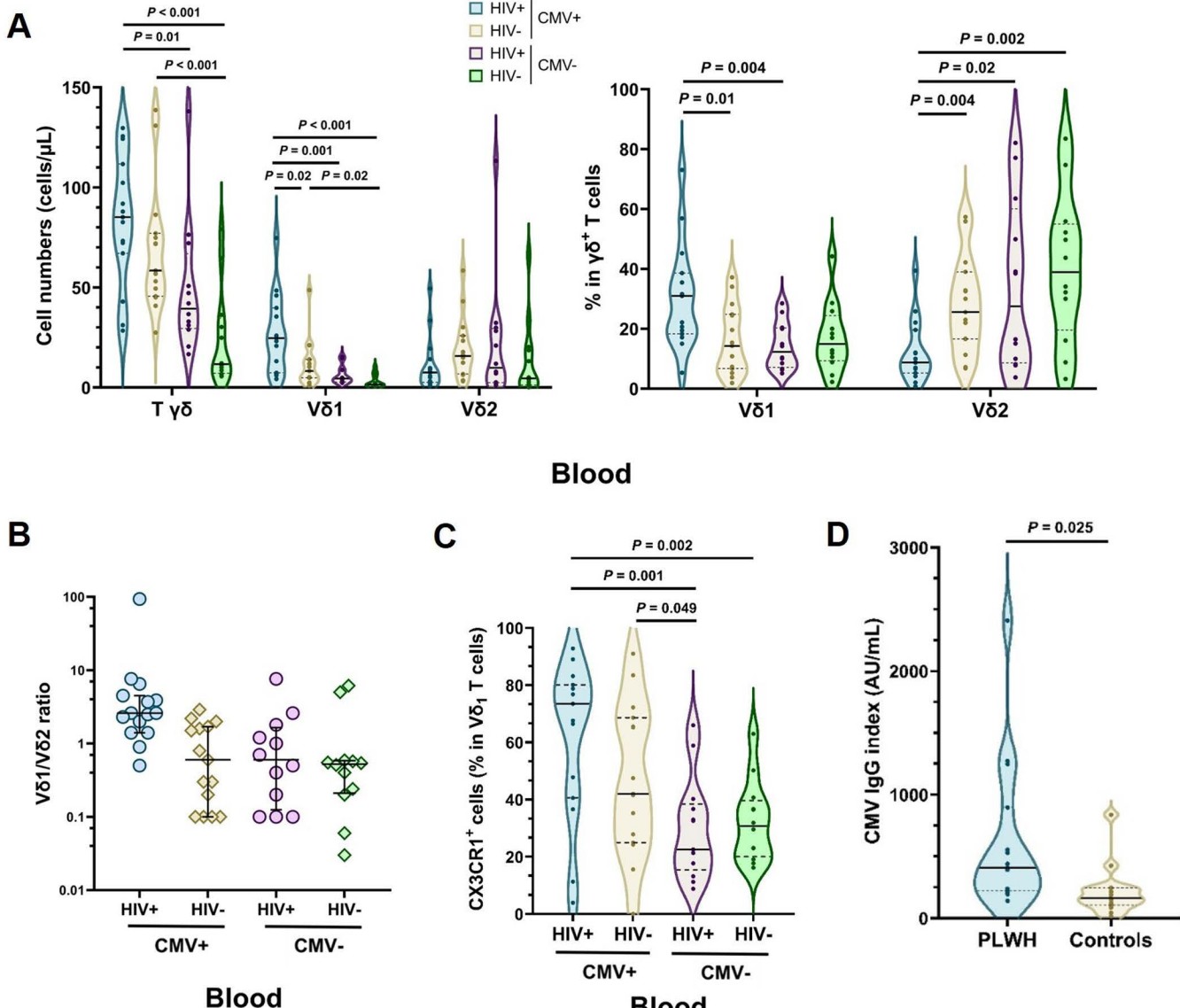

**Fig 6. CMV is associated with Vδ₁ T cells expansion in PLWH on ART.** (A) Violin plots of the number of γδ⁺ T cells and of their Vδ₁ and Vδ₂ subsets, and the frequencies of Vδ₁ and Vδ₂ subsets among γδ⁺ T cells in the blood according to HIV and CMV infection status (n = 15 HIV⁺ CMV⁺/ n = 15 HIV⁻ CMV⁺/ n = 12 HIV⁺ CMV⁻/ n = 12 HIV⁻ CMV⁻ individuals). Comparisons were made using Welch's t-test. The violin plots show the median (solid bar) and the first and third quartiles (dashed bars). (B) Vδ₁/Vδ₂ subset ratio among γδ⁺ T cells in the blood according to HIV and CMV infection status. (C) Violin plots of CX3CR1⁺ cell frequencies among Vδ₁ T cells in the blood according to HIV and CMV infection status. Comparisons were made using Welch's t-test. (D) Violin plots of CMV IgG index in HIV⁺ CMV⁺ (n = 15) and HIV⁻ CMV⁺ individuals (n = 15). Comparisons were made using Welch's t-test. See also S8 Fig for plasma inflammatory biomarkers comparisons between the same groups.

several bacteria in PLWH, particularly in the Firmicutes phylum and the *Staphylococcus* genus (Fig 7A). The blood-translocated microbiota also showed a higher abundance of several bacteria in PLWH, including *Candidatus pelagibacter, Arcobacter, Lactobacillus*, while few others were relatively more abundant in controls (Fig 7B).

**Cytotoxic CX3CR1$^+$ TEMRA Vδ$_1$ T cell expansion is associated with the interplay of CMV, microbiota changes, and HIV-1 persistence**

Given the observed shift in translocated and mucosal microbiota in PLWH, and the critical link established between CMV infection and expansion of circulating CX3CR1$^+$ Vδ$_1$ T cells, we sought a potential interplay between microbiota, CMV residual replication, and Vδ$_1$ T cell phenotype. The anti-CMV IgG index, used as a surrogate for residual CMV replication, was positively correlated with the frequency of some bacterial genera in the duodenal microbiota as well as with Vδ$_1$ cytotoxic functions (GzmB$^+$Perforin$^+$ cell frequency among both duodenal and circulating Vδ$_1$ T cells, Fig 8A, C). The frequency of these bacteria in the duodenal microbiota was also positively correlated with circulating CX3CR1$^+$ Vδ$_1$ frequency (e.g., *Porphyromonas* relative abundance in duodenum, Spearman's ρ = 0.71; Fig 8C) and also with circulating Vδ$_1$ cytotoxic functions (e.g., *Porphyromonas* frequency in duodenum, Spearman's ρ = 0.65). The bacteria higher in the translocated microbiota were mainly positively correlated with the frequency and cytotoxicity profile of circulating CX3CR1$^+$ TEMRA Vδ$_1$ cells. In contrast, the relative abundance of the genus *Pelomonas* in the translocated microbiota was associated with a less abundant TEMRA and CX3CR1$^+$ phenotype and a less activatory profile (NKG2C$^+$) of circulating Vδ$_1$ T cells (Fig 8A).

To rule out potential confounding biases related to systemic inflammation caused by CMV co-infection and/or changes in the microbiota, we measured inflammatory biomarkers in the plasma of PLWH (who were all receiving sustained, effective ART) and controls, both with and without CMV co-infection (n = 15 CMV-seropositive PLWH, n = 15 CMV-seropositive HIV-seronegative controls, n = 12 CMV-seronegative PLWH and n = 12 CMV-seronegative HIV-seronegative controls) (S8 Fig). IL-6, TNF-α and IP-10 (CXCL10, ligand of CXCR3) were higher in PLWH than in HIV-seronegative controls. There were no significant changes in IL-1β or IL-8. We did not observe an increase in sCD14 in PLWH, despite previous reports describing such an increase. This was confirmed to be due to insufficient statistical power in our study population, as plasma sCD14 was significantly increased when assessed in the entire ANRS EP61 GALT cohort (n = 42 PLWH and 42 controls). There were not significant changes in the inflammatory biomarkers related to CMV status.

We then investigated whether these inflammatory biomarkers were associated with either the CMV IgG index or the relative abundance of bacteria previously linked to Vδ$_1$ T cell expansion. The CMV IgG index did not correlate with any of the assessed inflammatory biomarkers. Regarding bacterial triggers, there was no overall positive correlation between microbiota bacterial genera and plasma inflammatory biomarkers, at the exception of a positive tendency between plasma levels of IL-8 and TNF-α and the frequency of *Pelomonas* DNA in blood (S9 Fig). However, *Pelomonas* frequency was negatively correlated with Vδ$_1$ changes in the blood (Fig 8A). Therefore, neither occult CMV replication nor bacterial changes associated with Vδ$_1$ T cell expansion seemed to be significant triggers of systemic inflammation in PLWH.

Furthermore, inflammatory plasma biomarkers did not correlate with the frequency or phenotypic changes of Vδ$_1$ T cells observed in PLWH, except for a positive association between IL-6 plasma levels and the Vδ$_1$/Vδ$_2$ ratio. However, this association is not supported by a correlation between IL-6 levels and an increase in circulating Vδ$_1$ T cell frequency (ρ = 0.003, P = 0.99), but rather by a correlation with a reduction in Vδ$_2$ T cell frequency (ρ = -0.60, P = 0.02) (S9 Fig).

Thus, the changes observed in Vδ$_1$ T cells in PLWH did not seem to be triggered by non-specific systemic inflammation.

Finally, we examined the relationship between the phenotype of Vδ$_1$ T cells in PLWH and the HIV-1 reservoir within the same subjects in both the blood and intestinal compartments, specifically the intact fraction of HIV-1 DNA and the cell-associated residual HIV-1 RNA. The mean frequency of intact HIV-1 DNA in the intestine (measured in 3 segments: duodenum, ileum and colon) was associated with a decreased diversity (*i.e.,* clonal expansion) of circulating Vδ$_1$ T cells. The cytotoxic functions (frequency of GzmB$^+$Perforin$^+$ cells) of circulating CX3CR1$^+$ TEMRA Vδ$_1$ T cells were negatively correlated with the frequency of intact proviral DNA in PBMC (Spearman's ρ = -0.54), and CX3CR1$^+$ Vδ$_1$ T cells were also

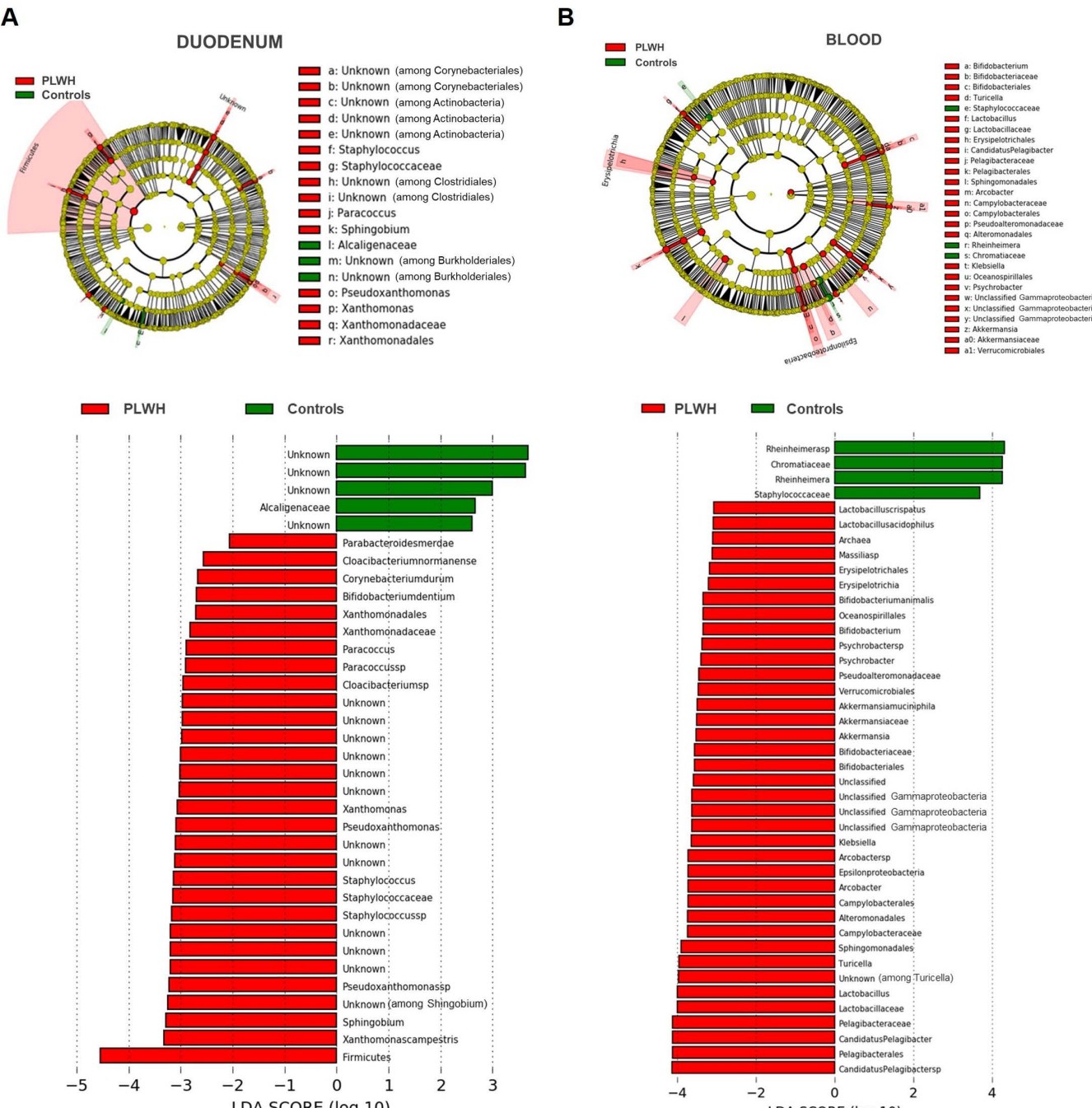

**Fig 7. PLWH show changes in duodenal mucosal and blood translocated microbiota.** Cladogram of phylogenetic relationships of bacterial lineages associated with HIV status and histogram of the linear discriminant analysis (LDA) scores for differentially abundant bacterial phyla, classes, orders, families and genera between PLWH and HIV seronegative controls. The taxonomic level in the cladograms is represented by phylum in the inner ring and genus in the outer ring, with each circle representing a taxon within that level. LDA scores represent the size and ranking of each differentially abundant taxon. Positive values (in green) represent higher abundance in HIV seronegative controls and negative values (in red) represent higher abundance in PLWH. (A) Duodenal biopsies (n = 24 PLWH and n = 25 HIV seronegative controls). LDA scores shown are *P < 0.05; (B) Blood samples (n = 33 PLWH and n = 39 HIV seronegative controls). LDA scores have threshold > 3 and **P < 0.01. See also S7 Fig for principal component analysis and diversity indices.

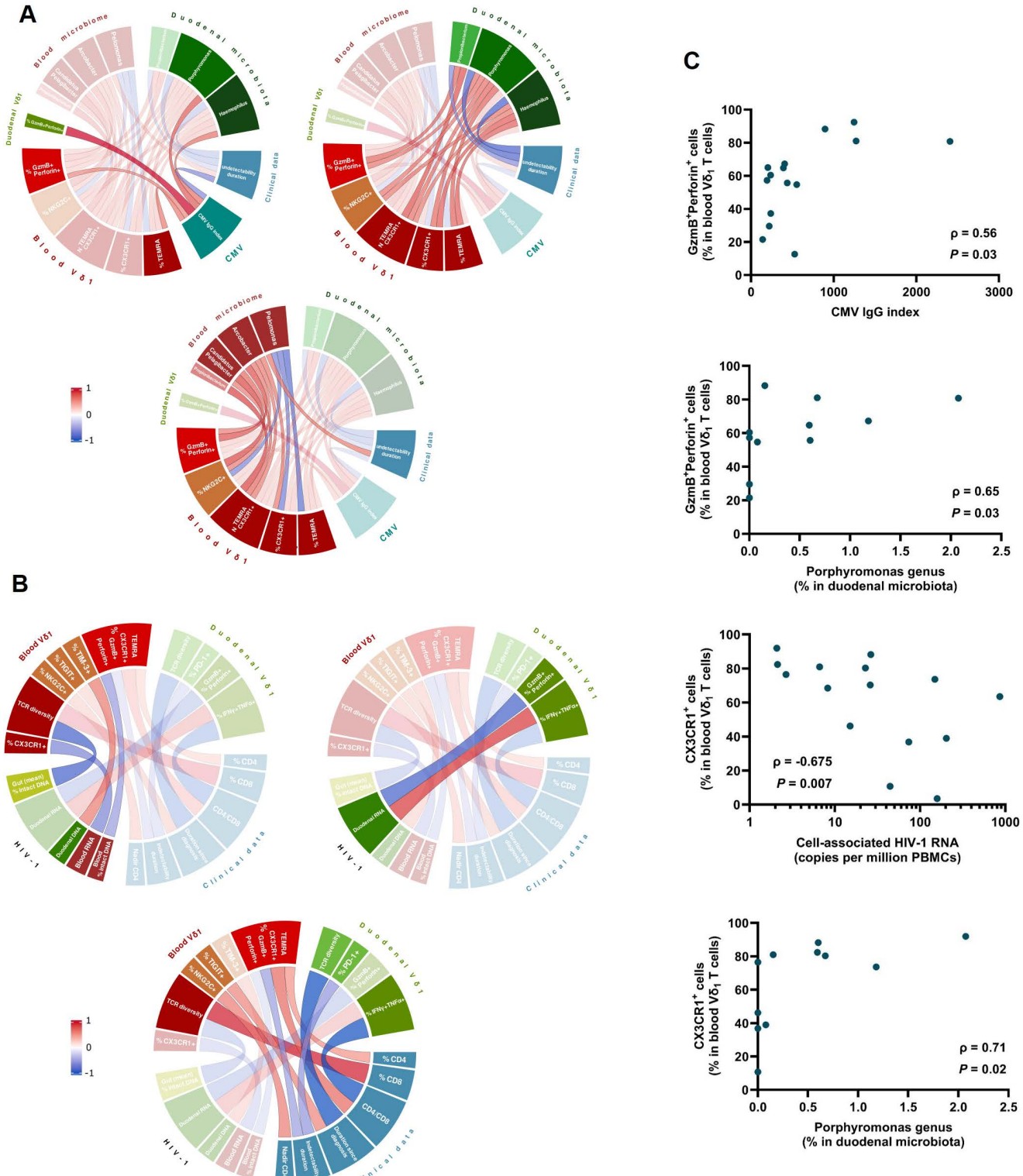

**Fig 8. Cytotoxic CX3CR1+ TEMRA Vδ₁ T cell expansion is associated with the interplay of CMV, microbiota changes and HIV-1 persistence.**
(A) Chord diagrams of the associations in PLWH between (i) CMV IgG index, changes in (ii) duodenal microbiota (relative abundance of some Proteo-
bacteria: *Haemophilus*; Bacteroidetes: *Porphyromonas*; and Actinobacteria: *Propionibacterium*), (iii) the blood microbiota (relative abundance of some

Proteobacteria: *Candidatus pelagibacter, Arcobacter, Pelomonas*; and Actinobacteria: *Propionibacterium*), and the phenotype of circulating and duodenal Vδ$_1$ T cells (CX3CR1$^+$ TEMRA Vδ$_1$ T cells and their cytotoxic or activating profile). Color scale indicates Spearman's correlation coefficient (*P* < 0.05). Positive correlations are shown in red and negative correlations are shown in blue. (B) Chord diagrams of the associations between the phenotype of circulating and duodenal Vδ$_1$ T cells (CX3CR1$^+$ TEMRA Vδ$_1$ T cells and their cytotoxic, activating or exhausted profile) and their clonal expansion (inverse of TCR diversity), virological parameters of the HIV-1 reservoir (frequency of intact proviral DNA among total HIV-1 DNA in (i) PMBC and (ii) gut; cell-associated HIV-1 RNA in PBMC and duodenal mucosa) and (iii) clinico-biological parameters of PLWH (percentage of CD4$^+$ and CD8$^+$ T cells, CD4/CD8 ratio, and nadir of CD4$^+$ T cell count in blood; duration between HIV-1 diagnosis and enrollment; duration of consecutive undetectable HIV viral load prior to enrollment). Color scale indicates Spearman's correlation coefficient (*P* < 0.05). Positive correlations are shown in red and negative correlations are shown in blue. (C) Dot plots of correlations between the frequency of circulating cytotoxic (GzmB$^+$Perforin$^+$) Vδ1 cells and (first panel) the CMV IgG index (n = 15 PLWH), and (second panel) *Porphyromonas* relative abundance in the duodenal mucosa (n = 11 PLWH); correlations between the frequency of circulating CX3CR1$^+$ Vδ$_1$ T cells and (third panel) cell-associated HIV-1 RNA in PBMC (copies per million of PBMC, n = 15), and (fourth panel) *Porphyromonas* relative abundance in the duodenal mucosa (n = 11 PLWH). Spearman's correlation coefficients (ρ) and P-values are shown. See also S9 Fig for correlations with plasma inflammatory biomarkers.

negatively correlated with residual cell-associated HIV-1 RNA in PBMC (n = 15, Spearman's ρ = -0.68, *P* = 0.007) (Fig 8B, C). Consistently, cytotoxic functions of duodenal Vδ$_1$ T cells (frequency of GzmB$^+$Perforin$^+$ cells) seemed to be negatively correlated with residual HIV-1 RNA in the duodenum (S10 Fig). In contrast, duodenal Vδ$_1$ T cells producing pro-inflammatory cytokines (frequency of IFN-γ$^+$TNF-α$^+$ cells), most of which do not express CX3CR1 and do not have cytotoxic properties, seemed to be positively correlated with residual HIV-1 RNA in the duodenum (S10 Fig). Taken together, these results suggest an antiviral potential of cytotoxic CX3CR1$^+$ Vδ$_1$ T cells against HIV-1-infected cells. Consistent with a potentially favorable profile of circulating cytotoxic CX3CR1$^+$ TEMRA Vδ$_1$ T cells, they were also positively correlated with CD4$^+$ T cell frequency and CD4/CD8 T cell ratio in the blood of PLWH on ART (n = 15, Spearman's ρ = 0.70, *P* = 0.048; S10 Fig).

## Discussion

Here we show that the increase in circulating Vδ$_1$ T cells observed in PLWH on ART is mostly a clonal expansion of terminally differentiated effector memory cells expressing the chemokine receptor CX3CR1 and exhibiting a functional profile characterized by high cytotoxicity but low cytokine production. Since Vδ$_1$ T cells are normally located primarily at mucosal barriers, and since we found evidence for trafficking of Vδ$_1$ cells with the gut, we suggest that the gut is likely to be an important trigger site for Vδ$_1$ T cell activation in PLWH. Our data highlight the critical link between occult CMV replication and Vδ$_1$ T cell expansion in PLWH. In addition to this key CMV-induced trigger, species-level changes within the duodenal microbiota composition and within the blood-translocated microbiota in PLWH are also associated with Vδ$_1$ T cell phenotypic changes. The association between the expansion of cytotoxic Vδ$_1$ T cells and the reduction of residual HIV-1 RNA in both blood and duodenal mucosa suggests that these cells may contribute to HIV-1 infection control. Taken together, our results unravel a complex interplay between CMV, bacterial microbiota and the immunological response of Vδ$_1$ T cells to HIV-1 infection.

The expansion of circulating Vδ$_1$ T cells, especially CX3CR1$^+$ cells, is associated with CMV status and is likely driven by occult CMV replication in PLWH despite being on effective ART with restored circulating CD4$^+$ T cell counts and undetectable CMV DNA in their blood. Residual CMV replication in PLWH on ART has been clearly established as a persistent driver of immune activation in previous studies and has been associated with inflammatory markers, CD8$^+$ T cell expansion, lower CD4/CD8 T cell ratio, occurrence of cardiovascular events, and microbial translocation [11,36–41]. Interestingly, PLWH are more frequently co-infected with CMV, with higher incidence and prevalence than HIV-seronegative individuals, although the reason remains unclear [42]. Analysis of blood samples from CMV-seronegative PLWH confirmed that Vδ$_1$ T cell expansion is associated with CMV serologic status. CMV IgG index has been suggested as a surrogate marker of residual CMV replication and was significantly higher in PLWH than in HIV-seronegative controls in our cohort [43–46]. While CD4$^+$ T cell deficiency is a known factor responsible for CMV replication in immunocompromised individuals, CMV viremia is almost always undetectable in PLWH on ART when their blood CD4$^+$ T cell counts are restored. We

hypothesize that persistent depletion of mucosal CD4$^+$ T cells in PLWH may contribute to tissular occult CMV replication even on ART with high blood CD4$^+$ T cell counts [3,4]. In settings other than HIV infection, such as common variable immunodeficiency or renal transplantation, CMV has also been described as a major driver of V$\delta_1$ T cell clonal expansion [21,47]. However, we also show significant correlations between duodenal mucosal microbiota composition and both CMV IgG index and V$\delta_1$ T cell phenotype. In the context of gut barrier dysfunction in PLWH, it is also possible that CMV may be a confounding factor between mucosal microbiota changes and V$\delta_1$ T cell activation in the gut. CMV IgG index has been associated with gut epithelial damage and an increase in microbial translocation markers, suggesting that CMV mucosal replication in the gut could reshape the local microbiota, which in turn could trigger an adaptive response by the V$\delta_1$ subset [41,48]. Indeed, we found changes in both the duodenal and blood-translocated microbiota in PLWH, in global diversity as well as species-specific changes. The specificity of the expanded V$\delta_1$ effector cell clones against CMV, some bacterial species or HIV-1 was not demonstrated in our study. Due to the lack of a TRDV1 CDR3 specificity database, we were unable to link the clonotype sequences to a specific pathogen, nor do we have sufficient numbers of cells to expand the clones ex vivo to assess their antiviral properties. Thus, the precise triggers and targets of expanded V$\delta_1$ T cells in PLWH could not be formally established here, but our data clearly highlight the key role of CMV co-infection in PLWH. Systemic inflammation does not appear to be a non-specific confounding factor, as plasma inflammatory biomarkers were not significantly associated with the expansion of CX3CR1$^+$ V$\delta_1$ T cells.

The hypothesis of intestinal activation of V$\delta_1$ T cells in PLWH is supported by the increased expression of the activator receptor NKG2C in V$\delta_1$ IEL, which correlated with the V$\delta_1$/V$\delta_2$ ratio in the blood. In addition, increased levels of circulating V$\delta_1$ T cells expressing both β7 integrin and CD103 (αE integrin), which are markers of intestinal residence, may indicate increased trafficking of V$\delta_1$ T cells with the gut in PLWH. Furthermore, the major circulating clonotype from one subject was also found in the duodenum of the same subject, supporting the idea of the recirculation of intestinal V$\delta_1$ T cells secondary to mucosal stimulation. The number of V$\delta_1$ clonotypes obtained from duodenal epithelium was lower than from blood samples because endoscopic duodenal biopsies only allowed extraction of a limited number of IEL. Thus, it is likely that not all existing duodenal clonotypes were analyzed and additional clones shared between blood and duodenum may have been missed. In addition, our study did not include the examination of clonotypes from other tissues such as skin or lung, which may also be sites of activation and recirculation of V$\delta_1$ T cells in PLWH [49,50].

CX3CR1 is the receptor for CX3CL1, also known as fractalkine, a chemokine expressed in endothelial and epithelial tissues, including the intestine. CX3CR1 appears to be expressed predominantly by CD27$^-$ effector T cells, both αβ and γδ T cells. A vascular tropism and thus a potential role of CX3CR1$^+$ T cells in endothelial immune surveillance has been described [17]. Increased CX3CR1 expression on CD4$^+$ and CD8$^+$ T cells in PLWH has been associated with CMV co-infection, with an enrichment of CMV-specific populations, some of which target the vascular endothelium [51–54]. Mirroring CX3CR1$^+$ CD8$^+$ T cells, CX3CR1$^+$ V$\delta_1$ T cells have enhanced cytotoxic functions driven by a similar transcriptional programming [55,56]. Our single-cell transcriptomic analyses allowed us to decipher more precisely the effector transcriptional profile of V$\delta_1$ T cells in PLWH, unraveling the loss of naive V$\delta_1$ T cells combined with the expansion of two distinct effector subsets in these individuals. Both effector subsets expanded in PLWH expressed high levels of CX3CR1 and the T-bet transcription factor, enabling important cytotoxic functions with high transcription of the *GZMB* and *GNLY* genes in particular, as well as high levels of the β7 integrin, suggesting a link to the gut. These two effector cell subsets could be distinguished in particular by their expression levels of TOX, which promotes the exhaustion pathway. Expanded V$\delta_1$ T cells in PLWH globally expressed higher levels of activation/exhaustion markers, particularly TIGIT and TIM-3, compared to HIV-seronegative controls. In contrast to the effector subsets expanded in PLWH, those reduced in PLWH were characterized by higher expression of Eomes and pro-inflammatory cytokine production such as IFN-γ [57]. Thus, CX3CR1$^+$ V$\delta_1$ T cells in PLWH were shifted toward a more cytotoxic effector profile rather than pro-inflammatory cytokine production. Cytotoxic CX3CR1$^+$ V$\delta_1$ T cells maintained a high expression of the transcription factor T-bet. This is similar to the CX3CR1$^{hi}$ CD8 T cell subset, which produces lower levels of IFN-γ and TNF-α than the CX3CR1$^{int}$ subset [58–60].

HIV-1 may also be a trigger for Vδ$_1$ T cell activation, as the mean frequency of intact HIV-1 DNA in the gut was associated with clonal expansion of circulating Vδ$_1$ T cells. In addition to being a possible trigger, HIV-1 may also be a target for Vδ$_1$ T cells, as the frequency of cytotoxic, but not of IFN-γ-producing, Vδ$_1$ T cells appeared to be negatively correlated with residual HIV-1 RNA in both the blood and duodenal compartments, suggesting a possible role for cytotoxic Vδ$_1$ T cells against reservoir cells with residual HIV-1 RNA transcription. However, the cytotoxic functions of Vδ$_1$ T cells were attenuated in the duodenal epithelium, suggesting that inhibitory factors in the tissue microenvironment may regulate the effector functions of tissue-resident T cells, and thus contribute to HIV-1 persistence in tissues of PLWH on ART [61–63].

In conclusion, we found that the inversion of the Vδ$_1$/Vδ$_2$ ratio of γδ T cell subsets in the blood of PLWH on ART is due to the clonal expansion of cytotoxic CX3CR1$^+$ terminally differentiated effector memory Vδ$_1$ T cells, some of which exhibit trafficking with the gut. Vδ$_1$ T cell expansion appears to be mostly associated with persistent occult CMV replication in PLWH, even on effective ART, but also with changes in the duodenal mucosal and translocated microbiota, which have themselves been reshaped by HIV-1 infection, making the interplay between CMV, bacterial microbiota and HIV-1 key triggers for Vδ$_1$ T cell activation. These results highlight the importance of CMV co-infection in the chronic immune activation that persists in PLWH, even on ART. CX3CR1$^+$ effector Vδ$_1$ T cells may have both deleterious and beneficial effects in this setting, with their trafficking to the vascular endothelium possibly contributing to cardiovascular events, while their cytotoxic properties appear to be associated with HIV-1 replication control in our work.

## Materials and methods

### Ethic statement

The study was approved by the Institutional Review Board Comité de Protection des Personnes Sud-Ouest et Outre-Mer I (IDRCB: 2016-A00823-48). Written informed consent was obtained from all the participants (trial registration number NCT02906137).

### Subjects and samples

The ANRS EP61 GALT study had enrolled 42 PLWH and 42 HIV-seronegative controls at the University Hospital of Toulouse, France, from 2017 to 2020. Inclusion and exclusion criteria were previously published [7]. Blood samples and intestinal endoscopic biopsies were obtained from these individuals during either upper endoscopy (duodenal biopsies), and/or lower endoscopy (ileal and colonic biopsies). Upper endoscopy was performed primarily for gastroesophageal reflux disease; lower endoscopy was performed primarily for colorectal cancer screening. All subjects were free of inflammatory or lymphoproliferative bowel disease on histopathologic examination. Here, we performed a substudy with blood and duodenal samples from 15 PLWH and 15 HIV-seronegative controls who were representative of the entire ANRS EP61 GALT cohort and seropositive for CMV IgG, matched for sex ratio and age. In addition, blood samples from 12 PLWH and 12 HIV-seronegative controls, all CMV seronegative, were selected as a control cohort. All PLWH initiated ART in the chronic stage of infection and had sustained undetectable plasma HIV-1 RNA on ART. Plasma HIV-1 RNA was quantified by Aptima HIV-1 Quant Dx (30 copies/mL cutoff, Hologic), and peripheral blood CD4$^+$ and CD8$^+$ T cells were measured by flow cytometry as part of routine monitoring.

### Isolation of intra-epithelial lymphocytes from the duodenal mucosa

Eight duodenal endoscopic punch biopsies were collected from each individual. Five biopsies were processed to harvest intraepithelial lymphocytes and the remaining 3 biopsies were kept for viral and bacterial DNA/RNA extraction, and paraffin and OCT inclusions, as previously published [7]. The 5 biopsies were collected in RPMI with 10% FBS before being pooled in a medium containing 1X HBSS without calcium, magnesium or phenol red, 10 mM HEPES, 1 mM DTT, and 10% FBS, and incubated for 1h at 37°C with 5% $CO_2$ and gentle shaking. The DTT allows the epithelial cells and IEL to detach

into the supernatant. The resulting supernatant was filtered through a 100 µm filter, and spun for 5 minutes at 400 g. The pellet was washed once with PBS and 2% FBS. EasySep EpCAM$^+$ magnetic sorting (Stemcell) was then performed. This step retained the epithelial cells. The negative fraction contained the IELs. Typically, 500,000–1 million IELs were collected per donor and cryopreserved in FBS/DMSO in liquid nitrogen. The IEL recovery rate in subsequent experiments was typically around 50%.

## Antibodies and flow cytometry

V$\delta_1$ and V$\delta_2$ T cell phenotypes in blood and duodenal mucosa were characterized using core panels to define cells of interest, supplemented by 5 different panels for extracellular expression of activation and exhaustion markers (panel 1) and of homing markers (panel 2); intracellular expression of cytotoxicity effectors (panel 3) and cytokines (panel 4); and intranuclear staining of transcription factors (panel 5). 50,000 IELs and 1 million PBMCs were used per panel for each donor. The markers, fluorochromes, clones and suppliers of the core and supplemental panels are listed in S2 Table. The number of PBMC and IEL samples, and the HIV and CMV status of the subjects studied are listed in S3 Table. Core panel results were the mean of the different experiments. The number of events acquired in the flow cytometry analyses for the main cell subsets is listed in S4 Table.

For the homing marker panel, staining with the chemokine receptor antibodies (anti-CCR6, CCR9, CXCR3, and CX3CR1) was performed with Brilliant Stain Buffer (BD Biosciences) at 37°C in 5% $CO_2$ for 2 hours, followed by washing and a second staining for 20 minutes with Brilliant Stain Buffer and the other antibodies in the panel. This prolonged staining allowed for improved detection of membrane-recycled chemokine receptors compared to a standard 20-minute staining. For the cytotoxicity effector, cytokines and transcription factors panels, cells were stained after an overnight culture in RPMI 10% FBS solution at 37°C in 5% $CO_2$. Intracellular expression of cytotoxicity effectors staining was performed without cell stimulation, while cytokines staining was performed after cell stimulation with PMA at 100ng/mL and ionomycin at 1000ng/mL for 6 hours, and the addition of Golgi stop and Golgi plug (BD Biosciences) after 2 hours of stimulation. Intracellular staining was performed after permeabilization with Cytofix/Cytoperm and Perm/Wash buffer (Fixation/Permeabilization kit, BD Biosciences). For intranuclear staining, permeabilization was performed with Foxp3/transcription factor staining buffer set (eBioscience, Invitrogen).

Flow cytometry experiments were performed on a BD Symphony A5 driven by the FACSDiva software (BD Biosciences). Instrument reproducibility was verified for each experiment according to an in-house quality control procedure using CS&T and 8-peak rainbow beads. In addition, the cytometer was standardized to ensure consistency of the results over time according to BD Biosciences application settings. Fluorescence Minus One (FMO) controls were used to improve gating positioning. Analyses were performed using FlowJo software (v10.9.0) and OMIQ software from Dotmatics (https://www.dotmatics.com/solutions/omiq) for Opt-SNE analysis and cluster identification using the FlowSOM algorithm. Opt-SNE is a variant of the t-SNE algorithm that features modifications to the Barnes-Hut implementation of t-SNE, including the ability to detect the rate of improvement of the KL divergence and then automatically stop the algorithm when it begins to suffer from diminishing returns in this metric [64]. Parameters used for dimensionality reduction included: CD27, CD45RA, CX3CR1, NKG2A, NKG2C, NKp30, NKp44, NKp46, LAG-3, PD-1, TIGIT and TIM-3.

## Cell sorting

For V$\delta_1$ repertoire analysis, cells were stained with LIVE/DEAD Fixable Blue Dead Cell Stain (ThermoFisher), anti-human CD3 BUV395 (UCHT1/BD), γδ TCR BB700 (11F2/BD), and V$\delta_1$ APC Vio770 (REA173/Miltenyi), passed through a 0.7µm cell strainer and sorted using a FACSARIA Fusion or a FACSARIA SORP (BD Biosciences) cell sorter. Sorted Vδ1 cells were placed in RNAlater (Sigma Aldrich) and frozen. Vδ1 cells were sorted from PBMC of 5 PLWH and 5 HIV seronegative controls, all CMV seropositive, and from duodenal IEL of the same 5 PLWH.

For $V\delta_1$ single-cell RNA sequencing, cells were processed identically except for additional staining with TotalSeq-B anti-human hashtags (LNH-94; 2M2/BioLegend) directed against CD298 and β2-microglobulin (6 different hashtags were used for each donor). Sorted $V\delta_1$ T cells were stored on ice and immediately processed for single-cell library generation. Vδ1 cells were sorted from PBMC of 6 PLWH and 6 controls, all CMV seropositive.

## Single cell RNA sequencing and analysis

Single-cell libraries were generated using the Chromium iX Single-Cell Instrument and Chromium Next GEM Single Cell 3′ Kit v3.1, Chip G Single Cell kit, Dual Index kit TT Set A and NT set A, and 3' Feature Barcode kit (10x Genomics). Briefly, sorted $V\delta_1$ T cells were immediately diluted if necessary to equilibrate cell concentrations between samples before being pooled in a 19,800 cell solution in the appropriate volume for sample loading. After generation of nanoliter-scale gel bead-in-emulsions, libraries for single cell RNA and hashing antibodies were constructed according to the manufacturer's recommendations. The obtained libraries had an average length of 353 bp and a concentration of 14.44nM. Sequencing was then performed on a full SP flow cell 100 cycles (2 lanes) on the Illumina NovaSeq 6000 instrument (Illumina, San Diego, USA) at the GeT-PlaGe site of the Genome and Trancriptome Core Facility (GeT, Genotoul, Toulouse, France), using the NovaSeq 6000 SP Reagent Kit v1.5 (100 cycles) and appropriate run settings (R1: 88 bp, R2: 28 pb, I1: 10, I2: 10), to target approximately 27,500 + 5,000 reads per cell with an estimated cell recovery rate of 50%.

Raw reads were processed using 10x Genomics *Cell Ranger* v.7.0.1 [65] with default parameters. Briefly, GEX reads with valid barcodes were aligned to the Homo sapiens GRCh38 reference genome and Ensembl annotation v.110, while CMO reads were aligned to the HTO reference sequences. Aligned reads with good mapping quality were then counted. Single cell RNA and CMO sequencing counts were analyzed using the R package Seurat v.5.0.1 for R version 4.3.2 [66]. For our data we select cells with at least 500 counts per cell, at least 500 expressed genes, a complexity greater than 0.8 and a mitochondrial gene ratio less than 0.2. After filtering to remove empty droplets and dying cells, only genes expressed in 10 or more cells were retained, resulting in a total of 23,974 unique genes. The hash detection step allowed cells to be associated with their corresponding tag and to identify untagged cells and doublet droplets that were removed [67]. Approximately 650 cells were analyzed per individual.

Raw counts were normalized. PCA was performed using the 3,000 more variable features. An integration step was performed using the CCAIntegration method and the UMAP was generated based on the first 30 PCs. Clustering was performed at 0.2 resolution using the Louvain algorithm with multilevel refinement and 50 iterations. A cluster of contaminating monocytes was identified and removed using the CD14, CCR2 and SELL gene expression. We identified 6 clusters and used *propeller* to find differences in cell type proportions between PLWH and controls [68]. Pseudobulk counts were generated by aggregating cell counts from each cluster and each patient. Differential expression analyses were performed using the R package DESeq2 v.1.42.1 [69] with default parameters. Differentially expressed genes were identified as genes with an adjusted *P*-value less than 0.05.

## TCR repertoire analysis

Sorted $V\delta_1$ T cells cells from PBMC and duodenal IEL (1,100–175,000 cells) were placed in RNAlater (Sigma Aldrich) and frozen. Deep, quantitative and unbiased amplification of TCRδ sequences was performed using dimer-avoided multiplex (dam)-PCR, followed by MiSeq sequencing (illumina) performed by iRepertoire, Inc. (Huntsville, USA). The sequencing data were error-corrected and V, D, and J gene usage and complementarity-determining region 3 (CDR3) sequences were identified and assigned. Tree maps show each unique CDR3 as a colored rectangle, with the size of each rectangle corresponding to the abundance of each CDR3s within the repertoire. The diversity index was calculated using iRweb tools (iRepertoire, Inc, Huntsville, AL, USA). The higher the diversity of the sample, the closer the overall diversity of the sample fits a diversity model where each clonotype receives equivalent reads. Cleaned nucleotide sequence alignments were used to calculate Hill's numbers, which characterize abundance-based species complexity [70]. Hill's numbers

represent a set of numbers of different order (q-value) for which increasing values make the measure of complexity progressively less sensitive to rare haplotypes and at infinity, only the frequency of the dominant haplotype matters. We used Rényi's plots to order the diversity of *TRDV1* clonotypes [71].

## Bacterial 16S rRNA gene sequencing

DNA for bacterial 16S rRNA gene sequencing was obtained from intestinal tissue and whole blood by thorough lysis using SK38 glass and ceramic beads in a Precellys 24 homogenizer (Bertin Technologies) for 6 cycles of 30 seconds each at 6500 rpm with 1 minute breaks on ice. The 16S bacterial DNA V3–V4 regions were targeted with 357wf-785R primers and analyzed using MiSeq (RTLGenomics, Texas, USA). For each blood and duodenum sample, an average of 23,948 and 55,570 bacterial genome sequences were obtained, respectively. Bioinformatic filters were applied as described previously [72]. Cladograms and Linear Discriminant Analysis (LDA) scores were generated using the LDA Effect Size (LEfSe) module of the Huttenhower Lab Galaxy Server (http://galaxy.biobakery.org) [73]. For cladograms, the alpha value for the factorial Kruskal-Wallis test between classes and the alpha value for the pairwise Wilcoxon test between subclasses were set to 0.01, and the threshold for the logarithmic LDA score for discriminative features was set to 2.0. Principal Component Analysis (PCA) plots were generated and OUT-based diversity indices were calculated using Past 4.17c software [74].

## Quantification of HIV-1 DNA and RNA in PBMC and intestinal mucosa

Extraction of DNA and RNA from PBMC and intestinal mucosa and quantification of both total and intact HIV-1 DNA and HIV-1 RNA were performed as previously described [7]. Briefly, PBMC and intestinal biopsies were gently lysed with CK14 ceramic beads using a Precellys 24 homogenizer (Bertin Technologies) for a short time (10s twice) and at low speed (5000 rpm) to minimize DNA shearing and to obtain high molecular weight (HMW) DNA, which was isolated using the Gentra Puregene kit (Qiagen). RNA was isolated using the RNeasy micro kit with an on-column DNase treatment (Qiagen).

HIV-1 DNA (LTR region) was quantified by real-time PCR (LightCycler 480 system, Roche) from DNA extracted from PBMC and intestinal biopsies using the Generic HIV DNA Cell kit (Biocentric). Diploid cell equivalents were quantified by real-time PCR of the GAPDH gene. The frequency of intact HIV-1 proviruses in circulating and intestinal CD4$^+$ T cells was determined by the intact proviral DNA assay (IPDA) as previously described [7,75]. Briefly, two multiplex ddPCRs were performed in parallel on a QX200 ddPCR system (Bio-Rad) to quantify intact and defective HIV-1 proviruses in the first ddPCR, and DNA shear and diploid cell equivalents in the second ddPCR. Four to eight replicates of ddPCR were performed for HIV-1 provirus quantification, each from 750 ng of HMW DNA. Duplicates of RPP30 ddPCR were performed, using 70 ng of HMW DNA. Replicate wells were pooled and results were analyzed using QuantaSoft software (1.6.6.0320).

RNA extracted from PBMC and intestinal biopsies was reverse transcribed into cDNA using the Superscript IV (ThermoFisher Scientific) with random hexamers (1.25 μM) and oligo(dT)20 primers (1.25 μM). Tissue and cell-associated HIV-1 RNA (LTR region) was quantified by real-time PCR (LightCycler 480 system, Roche) from the cDNA using the Generic HIV DNA Cell kit (Biocentric) and a standard curve derived from a sample with a known concentration of HIV-1 RNA. Results are reported as cell-associated HIV-1 RNA copies per million PBMC, and HIV-1 RNA copies per μg of total intestinal tissue RNA, for each intestinal segment and as a mean value for the different intestinal segments of a given subject.

## Immunofluorescence staining

Immunohistochemistry was performed on paraffin-embedded duodenal biopsies. Briefly, 4 μm tissue section slides were deparaffinized and antigen retrieval was performed in a pH 9 citrate solution for 20 minutes at 95°C. Non-specific binding was blocked with MAXblock Blocking Medium (Active Motif) for 2 hours at room temperature. Anti-γδ TCR (5A6.E9/Thermo Fisher), anti-V$\delta_1$ (TS8.2/Thermo Fisher), and anti-V$\delta_2$ (15D/Thermo Fisher) monoclonal mouse IgG1 anti-human

monoclonal antibodies were used at 1/50 dilution for primary staining overnight at 4°C. Samples were then washed in PBS-Tween 0.01% and incubated for 1 hour at room temperature in a dark chamber with a goat anti-mouse IgG1 secondary antibody coupled to Alexa Fluor 555 (A-32727/Thermo Fisher) at 1/500 dilution. Cell nuclei were stained with DAPI (homemade preparation). Slides were mounted with ProLong Glass Antifade Mountant (Invitrogen, Thermo Fisher). Data were acquired using a Zeiss Apotome.2 wild field microscope for cell counting and a Leica SP8-STED 3X confocal microscope. Cytoplasmic staining associated with a DAPI-stained nucleus was counted using an automated algorithm in ImageJ and then applied to the total area of each observed sample to obtain a count per area measure.

### Plasma inflammatory biomarkers

Plasma inflammatory biomarkers were measured by electrochemiluminescence using the V-plex human biomarkers 1 assay (IL-1β, IL-6, IL-8, TNFα) and the V-plex custom chemokine panel 1 (IP-10) assay, both from Meso Scale Discovery. Plates were read using a MESO QuickPlex SQ 120 reader. The data were analyzed using Methodical Mind Analysis software. For sCD14 measurement, a human CD14 ELISA Quantikine kit (R&D Systems) was used.

### Statistical analysis

Quantitative variables were compared using the Welch's t-test for unequal variances. The paired t-test was used to compare paired intra-individual samples from the blood and intestinal compartments. Correlations between quantitative variables were estimated using Spearman's rank correlation coefficients. All tests were 2-sided, and $P$ values <0.05 were considered statistically significant. Statistical analyses were performed using Stata SE 16.1. Graphs were generated using GraphPad Prism 10.0. Chord diagrams were generated using the Circlize package in R [76]. Statistical analyses used to analyze flow cytometry, single-cell RNAseq, TCR repertoire, and bacterial genome data are described in the corresponding sections.

## Supporting information

**S1 Table. Clinical characteristics of the study population.** Quantitative data are median (IQR). a Self-reported by the study participants, but none of them identified themselves as transgender. b Time on ART: time since the first antiretroviral therapy. c Duration of aviremia: time with continuous undetectable viral load prior to sampling.
(DOCX)

**S2 Table. Antibodies used in flow cytometry panels.**
(DOCX)

**S3 Table. Number of samples processed per cytometry panel.**
(DOCX)

**S4 Table. Mean number of events acquired in flow cytometry analyses for the main subsets.**
(DOCX)

**S1 Fig. Flow cytometry gating strategy for γδ T cells in IELs.** Flow cytometric analysis of γδ$^+$ T cells among CD3$^+$ T cells and Vδ$_1$ and Vδ$_2$ subsets in a PLWH duodenal sample.
(TIF)

**S2 Fig. Vδ$_1$ T cell numbers are not significantly altered in the duodenum of PLWH on ART.** (A) Representative immunofluorescence staining for Vδ$_1$ TCR (red) and DAPI (blue) in duodenal tissue from PLWH. Scale bars, 22µm. (B) Violin plots of the number of Vδ$_1$$^+$ T and Vδ$_2$$^+$ cells per surface area of duodenal tissue surface (n = 15 PLWH and 15 controls). Comparisons were made using Welch's t-test.
(TIF)

**S3 Fig. Chemokine receptors expression on blood Vδ1 T cells.** Violin plots of the chemokine receptors CXCR3, CCR6 and CCR9 frequencies among circulating Vδ1 T cells (n = 15 PLWH and 15 controls).
(TIF)

**S4 Fig. TEMRA γδ T cells show high expression of activation and exhaustion markers in PLWH on ART.** Each colored circle represents the proportion of the given marker expression among TEMRA (CD27$^-$CD45RA$^+$) Vδ$_1$ or Vδ$_2$ T cells as measured by flow cytometry (median for n = 15 PLWH and 15 controls). Overlay of colored circles represents co-expression of markers.
(TIF)

**S5 Fig Duodenal Vδ1 T cells express higher levels of Perforin$^+$GzmB$^-$ cells in PLWH on ART.** Dot plots of the frequencies of Perforin$^+$GzmB$^-$ among CX3CR1$^+$ Vδ$_1$ T cells in the IEL (n = 5 PLWH and 5 controls). Comparison was made using Welch's t-test.
(TIF)

**S6 Fig. Differentially expressed genes within Vδ$_1$ T cell clusters in PLWH on ART.** Volcano plots of pseudobulk differential gene expression between PLWH and controls (increased or decreased expression in PLWH vs. controls are shown in red and blue, respectively) for the total sequenced cells, clusters 0, 1 (increased frequency in PLWH), and 3 (decreased frequency in PLWH).
(TIF)

**S7 Fig. Differential microbiota profiling reveals increased diversity in blood and duodenum of PLWH on ART.** Principal component difference analysis and diversity indices between PLWH (red) and controls (green) in (A) blood samples (n = 33 PLWH and 39 controls); (B) duodenal biopsies (n = 24 PLWH and 25 controls). Two-way ANOVA followed by Benjamini, Krieger, and Yekutieli two-stage linear step-up procedure to correct for multiple comparisons by controlling for false discovery rate (<0.05). *P*-value of ** < 0.01, *** < 0.001, and **** < 0.0001.
(TIF)

**S8 Fig. Increase in inflammatory biomarkers is related to HIV status rather than CMV status.** (A) Dot plots of plasma concentration of IL-1β, IL-6, IL-8, TNFα, sCD14 and IP-10 according to HIV-1 status. (B) Violin plots of sCD14 plasma concentration in the entire ANRS EP61 GALT cohort (n = 42 PLWH and 42 controls). (C) Dot plots of plasma concentration of IL-1β, IL-6, IL-8, TNFα, sCD14, and IP-10 according to CMV status. (D) Violin plots of plasma concentration of IL-1β, IL-6, IL-8, TNFα, sCD14 and IP-10 according to both CMV and HIV-1 status. N = 15 CMV-seropositive PLWH (blue), n = 15 CMV-seropositive HIV-seronegative controls (brown), n = 12 CMV-seronegative PLWH (purple), and n = 12 CMV-seronegative HIV-seronegative controls (green). Bars are median and interquartile range. Comparisons were made using Welch's t-test.
(TIF)

**S9 Fig. Inflammatory biomarkers do not correlate with CX3CR1$^+$ Vδ$_1$ T cells expansion.** (A) Correlation matrix of plasma inflammatory biomarkers and (left panel) relative abundance of some bacteria genera in the blood and duodenum microbiota; and (right panel) the phenotype of circulating Vδ$_1$ T cells (n = 15 CMV-seropositive PLWH). Color scale indicates Spearman's correlation coefficient. *P*-values of * < 0.05, ** < 0.01, and *** < 0.001. (B) Correlations between plasma IL-6 and Vδ$_1$/Vδ$_2$ ratio, Vδ$_1$, and Vδ$_2$ T cell frequencies in blood. Spearman's correlation coefficients (ρ) and P-values are shown (n = 15 CMV-seropositive PLWH).
(TIF)

**S10 Fig. Cytotoxic CX3CR1$^+$ TEMRA Vδ$_1$ T cell expansion seems to be associated with reduced cell-associated HIV-1 RNA and higher T CD4 cell frequency in blood.** Correlations between cell-associated HIV-1 RNA

in the duodenum and (left upper panel) the frequency of duodenal cytotoxic (GzmB⁺Perforin⁺) Vδ1 cells, and (right upper panel) the frequency of duodenal IFNγ⁺TNFα⁺ V$\delta_1$ cells (n = 5 PLWH); correlations between the frequency of circulating cytotoxic (GzmB⁺Perforin⁺) Vδ1 cells and (left lower panel) the blood CD4/CD8 ratio, and (right lower panel) the frequency of CD4 T cells in the blood (n = 15 PLWH). Spearman's correlation coefficients (ρ) and P-values are shown. (TIF)

**S1 Data. Flow cytometry data, biomarkers, and CMV IgG Index.**
(XLSX)

## Acknowledgments

We thank the patients who participated in this study. We also thank S. Lagarrigue, A. Frelat, and C. Pomes for their help in monitoring the study; M-P. Panero, I. Da Silva, M. Pucelle and L. Staes for their help with the technical experiments. Flow cytometry experiments were performed at the INFINITy-INSERM UMR1291 core facility connected to the Toulouse Réseau Imagerie network, member of the France-BioImaging national infrastructure supported by the French National Research Agency (ANR-10-INBS-04). The authors thank V. Duplan-Eche, H. Garnier and F.-E. L'Faqihi for their help with flow cytometry; N. Jeanne for bioinformatics assistance; M. Zahm for transcriptomic bioinformatics analysis; A. Chaubet for help with single cell experiments; E. Lhuillier (GeT, Genotoul, Toulouse, France) for help with NGS experiments; ANEX-PLO US06 Purpan histopathology platform; and L. Lobjois for help with microscopy. Fig 1A was created using BioRender. Vellas, C. (2025) https://BioRender.com/e91f817 and https://BioRender.com/d39b132.

## Author contributions

**Conceptualization:** Pierre Delobel.

**Data curation:** Nived Collercandy, Camille Vellas, Manon Nayrac, Mary Requena.

**Formal analysis:** Nived Collercandy, Camille Vellas, Mary Requena, Matteo Serino, Pierre Delobel.

**Funding acquisition:** Pierre Delobel.

**Investigation:** Nived Collercandy, Camille Vellas, Manon Nayrac, Mary Requena, Thomas Richarme, Anne-Laure Iscache.

**Methodology:** Nived Collercandy, Pierre Delobel.

**Project administration:** Nived Collercandy, Pierre Delobel.

**Resources:** Karl Barange, Laurent Alric, Guillaume Martin-Blondel, Pierre Delobel.

**Software:** Justine Latour.

**Supervision:** Pierre Delobel.

**Validation:** Nived Collercandy, Camille Vellas, Mary Requena, Pierre Delobel.

**Visualization:** Nived Collercandy, Matteo Serino.

**Writing – original draft:** Nived Collercandy, Pierre Delobel.

**Writing – review & editing:** Nived Collercandy, Anne-Laure Iscache, Matteo Serino, Jacques Izopet, Pierre Delobel.

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
