## [Decision Letter · Decision Letter 0]

10 Jun 2025

Cytotoxic CX3CR1+ Vδ1 T cells clonally expand in an interplay of CMV, microbiota, and HIV-1 persistence in people on antiretroviral therapy

PLOS Pathogens

Dear Dr. Delobel,

Thank you for submitting your manuscript to PLOS Pathogens. After careful consideration, we feel that it has merit but does not fully meet PLOS Pathogens's publication criteria as it currently stands. Therefore, we invite you to submit a revised version of the manuscript that addresses the points raised during the review process.

Please submit your revised manuscript within 60 days Aug 09 2025 11:59PM. If you will need more time than this to complete your revisions, please reply to this message or contact the journal office at plospathogens@plos.org. Please include the following items when submitting your revised manuscript:

We look forward to receiving your revised manuscript.

Kind regards,

Jason M. Brenchley

Academic Editor

PLOS Pathogens

Richard Koup

Section Editor

PLOS Pathogens

Editor-in-Chief

PLOS Pathogens

orcid.org/0000-0003-2946-9497

Editor-in-Chief

PLOS Pathogens

orcid.org/0000-0002-7699-2064

**Additional Editor Comments :**

All reviewers found merit in the work. However they all recommend points which should be addressed (including some additional experimental data) before resubmission.

**Journal Requirements:**

At this stage, the following Authors/Authors require contributions: Nived Collercandy, Camille Vellas, Manon Nayrac, Mary Requena, Thomas Richarme, Anne-Laure Iscache, Justine Latour, Karl Barange, Laurent Alric, Guillaume Martin-Blondel, Matteo Serino, Jacques Izopet, and Pierre Delobel. Please ensure that the full contributions of each author are acknowledged in the "Add/Edit/Remove Authors" section of our submission form.

2) We noticed that you used the phrase 'data not shown' in the manuscript. We do not allow these references, as the PLOS data access policy requires that all data be either published with the manuscript or made available in a publicly accessible database. Please amend the supplementary material to include the referenced data or remove the references.

- ® on page: 18.

Potential Copyright Issues:

i) Figure 1a. Please confirm whether you drew the people images / clip-art within the figure panels by hand . If you did not draw the images, please provide (a) a link to the source of the images or icons and their license / terms of use; or (b) written permission from the copyright holder to publish the images or icons under our CC BY 4.0 license. Alternatively, you may replace the images with open source alternatives. See these open source resources you may use to replace images / clip-art:

6) In the online submission form, you indicated that "Data available upon request to corresponding author." All PLOS journals now require all data underlying the findings described in their manuscript to be freely available to other researchers, either

1. In a public repository

2. Within the manuscript itself

3. Uploaded as supplementary information.

7) Please amend your detailed Financial Disclosure statement. This is published with the article. It must therefore be completed in full sentences and contain the exact wording you wish to be published.

3) If any authors received a salary from any of your funders, please state which authors and which funders.

8) The following file is currently uploaded as file type 'Other', which is not viewable by the reviewers: STROBE_checklist_PPATH.docx. Please change the file type to 'Supporting Information' and include a legend in the manuscript if you wish it to be included in review.

**Reviewers' Comments:**

Reviewer's Responses to Questions

**Part I - Summary**

Reviewer #1: This manuscript by Collercandy et al. provides a comprehensive characterization of Vδ1 T cells in blood and duodenal IELs in the context of treated HIV-1 infection and CMV status utilizing phenotypic and functional markers, single-cell transcriptomics, TCR clonotype and microbiome analyses. The study is significant in its focus on duodenal IELs—an underexplored compartment in HIV infection—and offers valuable insights into the interplay between specific subsets of gut homing γδ T cells with CMV and gut bacteria during chronic HIV-1 infection and viral suppression with cART.

The finding that the clonal expansion of terminally differentiated effector memory Vδ1 T cells expressing the chemokine receptor CX3CR1 underlies the characteristic increase in PLWH on ART and represent a highly cytotoxic immune cell subset that likely contributes to control of HIV replication are significant. Additionally, the role of CMV replication and microbial stimulation on Vδ1 T cell effector functions add deeper understanding to the potential mechanisms that drive antiviral immune functions in this unique subset of immune cells at the interface of innate and adaptive immunity.

The manuscript is lucid, and the experimental approach is well-executed, and the discussion clearly states the implications, limitations and conclusions of the study.

Reviewer #2: The article is very informative. It addresses a critical knowledge gap in HIV immunology regarding the expansion of Vδ1 T cells in PLWH on ART. By integrating immunophenotyping, single-cell transcriptomics, TCR repertoire analysis, and microbiota sequencing, the authors provide a comprehensive picture of the behavior of Vδ1 T cells and their connection to CMV and microbiota.

The conclusions are well supported by the data:

• Flow cytometry and single-cell RNA sequencing confirm the expansion and cytotoxic profile of CX3CR1+ Vδ1-TEMRA cells.

• Clonal expansion and repertoire skew are demonstrated by TCR sequencing.

• Correlations between CMV serostatus, microbiota profiles, and Vδ1 T cell characteristics are supported by statistical analyses.

• The negative correlation between cytotoxic Vδ1 cells and residual HIV RNA supports a potential functional role in reservoir control.

Reviewer #3: This manuscript evaluates the changes in the Vdelta1 gd T cells that are associated with antiretroviral therapy in PLWH. The findings include an assessment of the phenotype of the Vd1 cells that make up the increased proportion of these gd T cells that appear in blood (CXCR1+/TEMRA), evidence for clonal expansion of these cells and that Vd1 cells have an increased cytotoxicity profile. Other findings include an association between CMV status and microbiota (both gut and translocated) and the Vd1 T cell levels.

This work represents an important contribution to our understanding of changes in Vd1 gd T cells that are associated with HIV infection and antiretroviral therapy, as well as identifying factors associated with these changes.

**Part II – Major Issues: Key Experiments Required for Acceptance**

Reviewer #1: None

Reviewer #2: Key concerns include:

1. Causality vs. correlation: The study relies heavily on correlations (between Vδ1 cytotoxicity and HIV-1 RNA), which limits conclusions about direct antiviral effects of Vδ1 T cells.

2. Limited functional assays: There is no direct functional evidence, which is of course explained by the limited cell count.

3. Scope of microbiota data: While associations with certain genera are interesting, mechanistic links between microbiota and immune phenotype remain speculative.

4. Limited sample size: The small cohort size (n = 15 PLWH and 15 controls in the main comparison) may limit statistical power, particularly in subanalyses (CMV stratification).

Reviewer #3: The one major issue I had with the manuscript is with regard to the associated factors that were identified (CMV status and microbiota). One might hypothesize that these factors have a common mechanism that is not explored in the manuscript, that has important implications for the conclusions. The hypothesis would suggest that these factors are altering the inflammation and immune activation environment of the host, and it is the resulting inflammatory mediators that are responsible for the altered Vd1 T cells. This is an important distinction particularly with regard to a mechanistic understanding as well as how one might therapeutically alter the Vd1 cells. As written the implication is that each contributing factor could be addressed separately, suppressing CMV replication or altering the microbiome or both (thereby addressing each of the ‘multiple triggers’ that the authors discuss). But if there is a common inflammatory/activation aspect to the elevation in Vd1 levels, then a potential treatment would impact the inflammation/activation environment Vd1 T cells. Indeed, a role for inflammation/immune activation is briefly touched on by the authors in referencing published work assessing residual CMV replication (one of the factors identified):

From Discussion: “Residual CMV replication in PLWH on ART has been clearly established as a persistent driver of immune activation in previous studies and has been associated with inflammatory markers, CD8+ T cell expansion, lower CD4/CD8 T cell ratio, occurrence of cardiovascular events, and microbial translocation (11,35–40). “

Due to this issue: My suggestion is that the authors undertake an additional experiment(s) to evaluate the levels of immune activation/inflammation in these patients, to identify any inflammatory/activation biomarkers associated with the changes in Vd1 levels. The type of assays undertaken might be guided by the manuscripts referenced in the quote above, or possibly other published studies. It is possible that sufficient information could be identified by assessing plasma/serum for levels of inflammation/immune activation associated cytokines/chemokines/biomarkers. The goal would be to determine if the factors that are associated with Vd1 cells (CMV status and microbial changes) are drivers of an inflammatory/immune activated environment. If this association proves to be correct, this provides an important mechanistic understanding. If there are no associations found that would also be important to know.

**Part III – Minor Issues: Editorial and Data Presentation Modifications**

Reviewer #1: Comments and suggestions to improve the manuscript are outlined below.

1. Materials and methods: A significant concern is the insufficient detail provided regarding the isolation of intraepithelial lymphocytes (IELs) from duodenal biopsies. The manuscript would benefit from a clear description of the number of pinch biopsies collected per subject, the specific processing protocol used for IEL isolation, and the typical cell yields obtained—particularly for the γδ T cell subsets. Additionally, the number of cells used per assay should be explicitly stated. Including a citation or reference to a previously validated protocol would strengthen the reproducibility and credibility of the methods.

2. To ensure rigor and reproducibility, the inclusion of representative flow cytometry plots illustrating the gating strategy for IEL γδ T cells is essential. This would help validate the identification and characterization of these subsets. Additionally, given that Vδ1 T cells and their phenotypic subsets can be low-frequency populations in gut tissues, it is important to provide a supplementary table detailing the number of events acquired for each subset presented in Figure 1. These additions would greatly enhance the transparency and interpretability of the data.

3. Line 94 – Clarification and Additional Context: The statement, “The Vδ1/Vδ2 ratio inversion seem to occur only at the chronic stage (30)”, requires grammatical correction—"seem" should be "seems." Additionally, it would strengthen the argument to reference supporting findings from the non-human primate (NHP) model of SIV infection under ART, as reported by Walker et al. (PMID: 33717202). Including this citation would provide further validation of the observed Vδ1/Vδ2 inversion and reinforce the relevance of the findings across species and disease models.

4. Figure 1 contains an excessive number of panels, which may hinder clarity and data interpretation. To improve readability and focus, it is recommended to move panels G–L into a new Figure 2. This separation would allow clearer emphasis on the NK cell and homing marker data, and better distinguish it from the earlier γδ T cell analyses.

5. The study design notes that ileal and colonic biopsies were also collected (lines 403–404), yet the manuscript focuses solely on duodenal Vδ1 T cell data. It would strengthen the study to include a comparison of Vδ1 T cell phenotype and function across these different gut regions. Such data could provide valuable insight into regional variation in mucosal immunity and help contextualize the findings within a broader immunological landscape of the gastrointestinal tract.

6. The data on CX3CR1 expression is compelling and suggests a potentially important role in the observed immune phenotype. However, it is unclear whether CX3CR1 was the only chemokine receptor analyzed. For a more comprehensive understanding of the trafficking and functional potential of these cells, it would be valuable to know if other chemokine or cytokine receptors—such as CXCR3, CCR5, or CCR6—were evaluated. Comparative expression data would enhance the interpretation of CX3CR1’s significance within the broader context of mucosal immune regulation.

7. Figure 1k presents a correlation between the blood Vδ1/Vδ2 ratio and the frequency of NKG2C⁺ Vδ1 T cells in IELs. However, the biological significance of this correlation is unclear—particularly given that the Vδ1/Vδ2 ratio inversion is a phenomenon observed predominantly in PLWH. Including both PLWH and healthy controls in the correlation analysis may obscure disease-specific associations, especially since healthy controls typically do not exhibit the inverted ratio. The authors should clarify the rationale for combining these groups and consider presenting stratified analyses to determine whether the correlation holds true within the PLWH cohort alone.

8. Line 178: The statement, “In contrast, circulating CX3CR1⁺ Vδ1 T cells had the lowest production of IFN-γ and TNF-α compared to CX3CR1⁻ cells,” appears counterintuitive, especially given the high T-bet expression in these cells. This discrepancy warrants further clarification. Could this be due to elevated baseline (unstimulated) cytokine levels, similar to the pattern observed for Granzyme B and Perforin? If so, it would be helpful to include baseline cytokine data or discuss this possibility explicitly to reconcile the apparent disconnect between transcription factor expression and cytokine output.

9. Line 196-198: The statement, “Cluster 4 had preserved but decreased CD27 expression, and also showed features of effector cells with the expression of cytotoxicity-associated genes,” would benefit from greater specificity. Please list the cytotoxicity-associated genes that characterize Cluster 4. Naming these genes will clarify the effector phenotype being described and allow readers to better interpret the functional identity of this cluster.

10. Line 230-231: “TRDV1 diversity was lower in PLWH blood, compared to both control blood and matched duodenal IEL, and remained lower for all values of q (representing the order of the diversity).” Please discuss the lower diversity of blood TRDV1 than IEL in PLWH. In the Fig. 4, it appears that blood still has higher diversity than IELs.

11. Subject 3124 appears to be an outlier in the control blood group, potentially showing evidence of clonal expansion. It would be valuable to discuss any unique clinical or immunological features of this individual, such as elevated CMV IgG titers or history of other infections that might drive clonal expansions. Including this information would help contextualize the observed data and clarify whether this subject represents a typical control or a special case.

12. Line 292-296: The manuscript mentions correlations involving duodenal IFN-γ⁺TNF-α⁺ Vδ1 T cells and HIV-1 RNA, as well as cytotoxic CX3CR1⁺ TEMRA Vδ1 T cells correlating positively with CD4⁺ T cell frequency and CD4/CD8 ratio, but the corresponding data are not shown. To enhance transparency and support these claims, the authors should include these correlation data—ideally as figures or supplementary material—allowing readers to evaluate the strength and significance of these associations.

13. The legend for Figure 4 states, “Vδ1 T cells from the duodenal IEL of 5 PLWH and PBMC from the same 5 PLWH and 5 HIV seronegative controls, all CMV seropositive, were used to sequence the TRDV1 chain repertoire.” However, according to the Methods section, TCR sequencing was performed on unsorted duodenal IELs and sorted Vδ1 T cells from blood of PLWH and HIV seronegative controls. This discrepancy should be corrected in the figure legend to accurately reflect the sample preparation and sorting strategy, ensuring clarity for readers.

14. Line 935-936: The statement, “TRDV1 chain in blood and Rényi’s plot of mean Hill’s numbers of order q = 0 to q = ∞, showing higher diversity in blood from PLWH (n=5) than in HIV seronegative controls (n=5),” appears inconsistent with the data. In the plot it appears that, except for subject #3124, HIV seronegative controls generally exhibit higher TRDV1 diversity in blood compared to PLWH. Please clarify.

Reviewer #2: The authors acknowledge that a causal relationship has not been established, particularly with regard to the antiviral function of Vδ1 T cells.

This study provides insights into how immune dysregulation persists in PLWH despite ART.

• It suggests that Vδ1 T cells could be used for immunotherapeutic strategies against the HIV reservoir.

• Provides further evidence of the interaction between microbiota, CMV, and immune cells in the formation of host immunity under chronic infection.

The tables and figures are well designed, informative, and appropriate for the data presented:

• The flow cytometry data are clearly presented.

• Single-cell and repertoire analyses are visualized using standardized and high-quality diagrams

• The figures include clear statistical annotations and appropriate comparisons

One potential weakness is that some figure captions are very dense and could be simplified for better comprehension.

Reviewer #3: Minor change suggested:

In Figure legend to figure 7, describe the chord diagrams in the order that they are displayed for ‘A’, right now CMV and Blood microbiota are switched. Specifically, the first chord diagram shown is of CMV, however that is (iii), the second is duodenal microbiota and that is in the proper place (ii) and the third is blood microbiota and that is (i).

PLOS authors have the option to publish the peer review history of their article (what does this mean? ). If published, this will include your full peer review and any attached files.

**Do you want your identity to be public for this peer review?** For information about this choice, including consent withdrawal, please see our Privacy Policy .

Reviewer #1: **Yes: ** Namita Rout

Reviewer #2: No

Reviewer #3: No

**Figure resubmission:**

**Reproducibility:**



---

## [Decision Letter · Decision Letter 1]

27 Aug 2025

Dear Prof Delobel,

We are pleased to inform you that your manuscript 'Cytotoxic CX3CR1+ Vδ1 T cells clonally expand in an interplay of CMV, microbiota, and HIV-1 persistence in people on antiretroviral therapy' has been provisionally accepted for publication in PLOS Pathogens.

Best regards,

Jason M. Brenchley

Academic Editor

PLOS Pathogens

Richard Koup

Section Editor

PLOS Pathogens

Sumita Bhaduri-McIntosh

Editor-in-Chief

PLOS Pathogens

orcid.org/0000-0003-2946-9497

Michael Malim

Editor-in-Chief

PLOS Pathogens

orcid.org/0000-0002-7699-2064

The reviewers are all in favor of publication, congrats!

Reviewer Comments (if any, and for reference):

Reviewer's Responses to Questions

**Part I - Summary**

Reviewer #1: (No Response)

Reviewer #2: The authors have adequately addressed the comments and suggestions in the revised version. I consider the revisions satisfactory and would recommend the manuscript for acceptance.

Reviewer #3: This manuscript provides important information that helps to unravel the underlying factors that drive the increase in Vd1 gamma delta T cells in PLWH. While some of the findings hinting to their function are correlative in nature, the findings still reveal key insights into this important T cell subset. The changes to the manuscript that were in response to the reviewer's critiques have resulted in clear improvements to this work.

**Part II – Major Issues: Key Experiments Required for Acceptance**

Reviewer #1: (No Response)

Reviewer #2: N.a.

Reviewer #3: (No Response)

**Part III – Minor Issues: Editorial and Data Presentation Modifications**

Reviewer #1: (No Response)

Reviewer #2: n.a.

Reviewer #3: (No Response)

PLOS authors have the option to publish the peer review history of their article (what does this mean? ). If published, this will include your full peer review and any attached files.

**Do you want your identity to be public for this peer review?** For information about this choice, including consent withdrawal, please see our Privacy Policy .

Reviewer #1: **Yes: ** Namita Rout

Reviewer #2: No

Reviewer #3: No

---

## [Editor Report · Acceptance letter]

Dear Prof Delobel,

We are delighted to inform you that your manuscript, "Cytotoxic CX3CR1+ Vδ1 T cells clonally expand in an interplay of CMV, microbiota, and HIV-1 persistence in people on antiretroviral therapy," has been formally accepted for publication in PLOS Pathogens.

Best regards,

Sumita Bhaduri-McIntosh

Editor-in-Chief

PLOS Pathogens

orcid.org/0000-0003-2946-9497

Michael Malim

Editor-in-Chief

PLOS Pathogens

orcid.org/0000-0002-7699-2064